# Triterpene Glycosides from the Far Eastern Sea Cucumber *Thyonidium (=Duasmodactyla) kurilensis* (Levin): The Structures, Cytotoxicities, and Biogenesis of Kurilosides A_3_, D_1_, G, H, I, I_1_, J, K, and K_1_

**DOI:** 10.3390/md19040187

**Published:** 2021-03-27

**Authors:** Alexandra S. Silchenko, Anatoly I. Kalinovsky, Sergey A. Avilov, Pelageya V. Andrijaschenko, Roman S. Popov, Pavel S. Dmitrenok, Ekaterina A. Chingizova, Vladimir I. Kalinin

**Affiliations:** G.B. Elyakov Pacific Institute of Bioorganic Chemistry, Far Eastern Branch of the Russian Academy of Sciences, Pr. 100-letya Vladivostoka 159, 690022 Vladivostok, Russia; silchenko_alexandra_s@piboc.dvo.ru (A.S.S.); kaaniv@piboc.dvo.ru (A.I.K.); avilov_sa@piboc.dvo.ru (S.A.A.); andrijashchenko_pv@piboc.dvo.ru (P.V.A.); popov_rs@piboc.dvo.ru (R.S.P.); paveldmt@piboc.dvo.ru (P.S.D.); chingizova_ea@piboc.dvo.ru (E.A.C.)

**Keywords:** *Thyonidium kurilensis*, triterpene glycosides, kurilosides, sea cucumber, cytotoxic activity

## Abstract

Nine new mono-, di-, and trisulfated triterpene penta- and hexaosides, kurilosides A_3_ (**1**), D_1_ (**2**), G (**3**), H (**4**), I (**5**), I_1_ (**6**), J (**7**), K (**8**), and K_1_ (**9**) and two desulfated derivatives, DS-kuriloside L (**10**), having a trisaccharide branched chain, and DS-kuriloside M (**11**), having hexa-*nor*-lanostane aglycone with a 7(8)-double bond, have been isolated from the Far-Eastern deep-water sea cucumber *Thyonidium (=Duasmodactyla) kurilensis* (Levin) and their structures were elucidated based on 2D NMR spectroscopy and HR-ESI mass-spectrometry. Five earlier unknown carbohydrate chains and two aglycones (having a 16*β*,(20S)-dihydroxy-fragment and a 16*β*-acetoxy,(20S)-hydroxy fragment) were found in these glycosides. All the glycosides **1**–**9** have a sulfate group at C-6 Glc, attached to C-4 Xyl1, while the positions of the other sulfate groups vary in different groups of kurilosides. The analysis of the structural features of the aglycones and the carbohydrate chains of all the glycosides of *T. kurilensis* showed their biogenetic relationships. Cytotoxic activities of the compounds **1**–**9** against mouse neuroblastoma Neuro 2a, normal epithelial JB-6 cells, and erythrocytes were studied. The highest cytotoxicity in the series was demonstrated by trisulfated hexaoside kuriloside H (**4**), having acetoxy-groups at C(16) and C(20), the latter one obviously compensated the absence of a side chain, essential for the membranolytic action of the glycosides. Kuriloside I_1_ (**6**), differing from **4** in the lacking of a terminal glucose residue in the bottom semi-chain, was slightly less active. The compounds **1**–**3**, **5**, and **8** did not demonstrate cytotoxic activity due to the presence of hydroxyl groups in their aglycones.

## 1. Introduction

The investigations of the triterpene glycosides from different species of sea cucumbers have a range of goals. Among them are the drug discoveries based on the promising candidates, demonstrating the target bioactivity [1,2,3,4,5,6], the solving of some taxonomic problems of the class Holothuroidea based on the specificity of the glycosides having characteristic structural peculiarities for the certain systematic groups [7,8,9,10], the ascertaining of biologic and ecologic functions of these metabolites [11,12,13,14,15], and the discovery of novel compounds, especially minor ones, that can be the “hot metabolites” clarifying the biosynthetic pathways of triterpene glycosides [16,17,18].

As a continuation of our investigation of glycoside composition of the sea cucumber *Thyonidium* (=*Duasmodactuyla*) *kurilensis* (Levin), we report herein the isolation and structure elucidation of nine glycosides, kurilosides A_3_ (**1**), D_1_ (**2**), G (**3**), H (**4**), I (**5**), I_1_ (**6**), J (**7**), K (**8**), and K_1_ (**9**) as well as two desulfated derivatives, DS-kuriloside L (**10**) and DS-kuriloside M (**11**). The animals were collected near Onekotan Island in the Sea of Okhotsk. The structures of the compounds **1**–**11** were established by the analyses of the ^1^H, ^13^C NMR, 1D TOCSY, and 2D NMR (^1^H,^1^H-COSY, HMBC, HSQC, ROESY) spectra as well as HR-ESI mass spectra. All the original spectra are presented in Appendix A. The hemolytic activities against mouse erythrocytes, cytotoxic activities against mouse neuroblastoma Neuro 2a, and normal epithelial JB-6 cells have been reported.

## 2. Results and Discussion

### 2.1. Structural Elucidation of the Glycosides

The concentrated ethanolic extract of the sea cucumber *Thyonidium (=Duasmodactyla) kurilensis* was chromatographed on a Polychrom-1 column (powdered Teflon, Biolar, Latvia). The glycosides were eluted with 50% EtOH and separated by repeated chromatography on Si gel columns using CHCl_3_/EtOH/H_2_O (100:100:17) and (100:125:25) as mobile phases to give five fractions (I–V). The glycosides **1**–**9** (Figure 1) were isolated as a result of subsequent HPLC of the fractions II–V on a reversed-phase semipreparative column Phenomenex Synergi Fusion RP (10 × 250 mm).

The molecular formula of kuriloside A_3_ (**1**) was determined to be C_54_H_87_O_29_SNa from the [M_Na_ − Na]^−^ ion peak at *m/z* 1231.5063 (calc. 1231.5059) in the (−)HR-ESI-MS. Kuriloside A_3_ (**1**) as well as the reported earlier kurilosides A, A_1_, and A_2_ [19] belong to the same group of glycosides, so these compounds have the identical monosulfated pentasaccharide chains that were confirmed by the coincidence of their ^1^H and ^13^C NMR spectra corresponding to the carbohydrate chains (Appendix A). The presence of five characteristic doublets at δ_H_ = 4.64–5.18 (*J* = 7.1–7.6 Hz), and corresponding signals of anomeric carbons at δ_C_ = 102.3–104.7 in the ^1^H and ^13^C NMR spectra of the carbohydrate part of **1** indicate the presence of a pentasaccharide chain and *β*-configurations of the glycosidic bonds. Monosaccharide composition of **1**, established by the analysis of the ^1^H,^1^H-COSY, HSQC, and 1D TOCSY spectra, includes one xylose (Xyl1), one quinovose (Qui2), two glucoses (Glc3 and Glc4), and one 3-O-methylglucose (MeGlc5) residue. The signal of C-6 Glc4 was observed at δ_C_ = 67.1 due to α-shifting effect of a sulfate group at this position. The positions of interglycosidic linkages were established by the ROESY and HMBC spectra (Appendix A). The analysis of NMR spectra of the aglycone part of **1** (Appendix A) indicated the presence of 22,23,24,25,26,27-hexa-*nor*-lanostane aglycone with a 16α-hydroxy,20-oxo-fragment and 9(11)-double bond due to the characteristic signals: (δ_C_ 149.0 (C-9) and 114.2 (C-11), δ_C_ = 71.1 (C-16) and δ_H_ = 5.40 (brt, *J* = 7.5 Hz, H-16), δ_C_ = 208.8 (C-20)). The ROE correlations H-16/H-15β and H-16/H-18 indicated a 16α-OH orientation in the aglycone of kuriloside A_3_ (**1**). 17αH-orientation, common for the sea cucumber glycosides, was deduced from the ROE-correlation H-17/H-32. The same aglycone was found earlier in kuriloside F [19].

The (−)ESI-MS/MS of **1** demonstrated the fragmentation of [M_Na_ − Na]^−^ ion at *m/z* 1231.5. The peaks of fragment ions were observed at *m/z* 1069.5 [M_Na_ – Na − C_6_H_10_O_5_ (Glc)]^−^, 1055.4 [M_Na_ – Na − C_7_H_12_O_5_(MeGlc)]^−^, 923.4 [M_Na_ − Na − C_6_H_10_O_5_(Glc) − C_6_H_10_O_4_(Qui)]^−^, 747.3 [M_Na_ − Na − C_6_H_10_O_5_(Glc) − C_6_H_10_O_4_(Qui) − C_7_H_12_O_5_(MeGlc)]^−^, 695.1 [M_Na_ −Na − C_24_H_37_O_3_(Agl) − C_6_H_10_O_5_ (Glc) − H]^−^, 565.1 [M_Na_ − Na − C_24_H_37_O_2_(Agl) − C_6_H_10_O_5_(Glc) − C_6_H_10_O_4_ (Qui) − H]^−^, 549.1 [M_Na_ − Na − C_24_H_37_O_3_(Agl) − C_6_H_10_O_5_(Glc) − C_6_H_10_O_4_(Qui) − H]^−^, 417.1 [M_Na_ – Na − C_24_H_37_O_3_(Agl) − C_6_H_10_O_5_(Glc) − C_6_H_10_O_4_(Qui) − C_5_H_8_O_4_(Xyl) − H]^−^, 241.0 [M_Na_ − Na − C_24_H_37_O_3_(Agl) − C_6_H_10_O_5_(Glc) − C_6_H_10_O_4_(Qui) − C_5_H_8_O_4_(Xyl) − C_7_H_12_O_5_(MeGlc) − H]^−^, corroborating the structure of kuriloside A_3_ (**1**).

All these data indicate that kuriloside A_3_ (**1**) is 3β-*O*-{β-d-glucopyranosyl-(1→4)-β-d-quinovopyranosyl-(1→2)-[3-*O*-methyl-β-d-glucopyranosyl-(1→3)-6-*O*-sodium sulfate-β-d-glucopyranosyl-(1→4)]-β-d-xylopyranosyl}-22,23,24,25,26,27-hexa-nor-16α-hydroxy,20-oxo-lanost-9(11)-ene.

The molecular formula of kuriloside D_1_ (**2**) was determined to be C_66_H_107_O_36_SNa from the [M_Na_ − Na]^−^ ion peak at *m/z* 1507.6291 (calc. 1507.6268) in the (−)HR-ESI-MS. The hexasaccharide monosulfated carbohydrate chain of **2** was identical to that of previously reported kuriloside D [19] since their ^1^H and ^13^C NMR spectra corresponding to the carbohydrate moieties were coincident (Appendix A). Actually, six signals of anomeric doublets at δ_H_ = 4.70–5.28 (d, *J* = 7.5–8.2 Hz) and corresponding signals of anomeric carbons at δ_C_ = 103.7–105.7 indicated the presence of a hexasaccharide chain in kuriloside D_1_ (**2**). The presence of xylose (Xyl1), quinovose (Qui2), three glucose (Glc3, Glc4, Glc5), and 3-O-methylglucose (MeGlc6) residues were deduced from the analysis of the ^1^H,^1^H-COSY, HSQC, and 1D TOCSY spectra of **2**. The positions of the interglycosidic linkages were elucidated based on the ROESY and HMBC correlations (Appendix A). The presence in the ^13^C NMR spectrum of kuriloside D_1_ (**2**) of the only signal of the *O*-methyl group at δ_C_ 60.5 and the upfield shift of the signal of C-3 Glc4 to δ_C_ 71.5 indicated the presence of a non-methylated terminal Glc4 residue. Analysis of the ^1^H and ^13^C NMR spectra of the aglycone part of **2** indicated the presence of a lanostane aglycone (the signals of lactone ring are absent and the signals of methyl group C-18 are observed at δ_C_ 16.9 and δ_H_ 1.30 (s, H-18) with normal side chain (30 carbons) and 9(11)-double bond (the signals at δ_C_ 149.0 (C-9), 114.9 (C-11), and δ_H_ 5.35 (brd, *J* = 6.2 Hz; H-11) (Table 1). The comparison of the ^13^C NMR spectra of **2** and kuriloside D showed their great similarity, except for the signals of the side chain from C-23 to C-27. Two strongly deshielded signals at δ_C_ 216.3 (C-16) and 217.6 (C-22) corresponded to carbonyl groups, whose positions were established on the base of the HMBC correlations H-15/C-16, H-21/C-22, H-23/C-22, and H-24/C-22. The signals of protons assigned to the methylene group adjacent to 22-oxo group were deshielded to δ_H_ 3.67 (dd, *J* = 10.6; 18.2 Hz; H-23a) and 3.43 (dt, *J* = 7.8; 18.2 Hz; H-23b) and correlated in the ^1^H,^1^H-COSY spectrum of **2** with one signal only at δ_H_ 2.27 (t, *J* = 7.8 Hz; H-24). These data, along with the deshielded signal of quaternary carbon at δ_C_ 69.0 (C-25) and the almost coinciding signals of methyl groups C-26 and C-27 (δ_C_ 30.0 and 29.5, δ_H_ 1.42 and 1.41, correspondingly), indicated the attachment of the hydroxy-group to C-25. Therefore, the side chain of kuriloside D_1_ (**2**) is characterized by the 22-oxo-25-hydroxy-fragment (Table 1).

The (−)ESI-MS/MS of **2** demonstrated the fragmentation of [M_Na_ − Na]^−^ ion at *m/z* 1507.6. The peaks of fragment ions were observed at *m/z* 1349.5 [M_Na_ − Na − C_8_H_15_O_3_ + H]^−^, corresponding to the loss of the aglycone fragment from C(20) to C(27), 1187.5 [M_Na_ − Na − C_8_H_15_O_3_ − C_6_H_10_O_5_(Glc) + H]^−^, 1025.4 [M_Na_ − Na − C_8_H_15_O_3_ − C_6_H_10_O_5_(Glc) − C_6_H_10_O_5_ (Glc) + H]^−^, 879.4 [M_Na_ − Na − C_8_H_15_O_3_ − C_6_H_10_O_5_(Glc) − C_6_H_10_O_5_(Glc) − C_6_H_10_O_4_(Qui) + H]^−^, 565.1 [M_Na_ − Na − C_30_H_47_O_4_(Agl) − C_6_H_10_O_5_(Glc) − C_6_H_10_O_5_(Glc) − C_6_H_10_O_4_(Qui) − H]^−^, 417.1 [M_Na_ − Na − C_30_H_47_O_5_(Agl) − C_6_H_10_O_5_(Glc) − C_6_H_10_O_5_(Glc) − C_6_H_10_O_4_(Qui) − C_5_H_8_O_4_(Xyl) − H]^−^, 241.0 [M_Na_ − Na − C_30_H_47_O_5_(Agl) − C_6_H_10_O_5_(Glc) − C_6_H_10_O_5_(Glc) − C_6_H_10_O_4_(Qui) − C_5_H_8_O_4_(Xyl) − C_7_H_12_O_5_(MeGlc) − H]^−^, corroborating the structure of kuriloside D_1_ (**2**).

All these data indicate that kuriloside D_1_ (**2**) is 3β-*O*-{β-d-glucopyranosyl-(1→3)-β-d-glucopyranosyl-(1→4)-β-d-quinovopyranosyl-(1→2)-[3-*O*-methyl-β-d-glucopyranosyl-(1→3)-6-*O*-sodium sulfate-β-d-glucopyranosyl-(1→4)]-β-d-xylopyranosyl}-16,22-dioxo-25-hydroxylanost-9(11)-ene.

The molecular formula of kuriloside G (**3**) was determined to be C_61_H_98_O_37_S_2_Na_2_ from the [M_2Na_ − Na]^−^ ion peak at *m/z* 1509.5102 (calc. 1509.5132) and the [M_2Na_ − 2Na]^2−^ ion-peak at *m/z* 743.2624 (calc. 743.2626) in the (−)HR-ESI-MS. In the ^1^H and ^13^C NMR spectra of the carbohydrate part of kuriloside G (**3**), six characteristic doublets at δ_H_ 4.65–5.19 (*J* = 7.0–8.1 Hz) and signals of anomeric carbons at δ_C_ 102.1–104.8, correlated with each anomeric proton by the HSQC spectrum, were indicative of a hexasaccharide chain and *β*-configurations of glycosidic bonds (Table 2). The signals of each monosaccharide unit were found as an isolated spin system based on the ^1^H,^1^H-COSY, and 1D TOCSY spectra of **3.** Further analysis of the HSQC and ROESY spectra resulted in the assigning of the monosaccharide residues as one xylose (Xyl1), one quinovose (Qui2), two glucoses (Glc3 and Glc5), and two 3-*O*-methylglucose (MeGlc4 and MeGlc6) residues.

The positions of interglycosidic linkages were established by the ROESY and HMBC spectra of **3** (Table 2) where the cross-peaks between H-1 Xyl1 and H-3 (C-3) of an aglycone, H-1 Qui2 and H-2 (C-2) Xyl1; H-1 Glc3 and H-4 (C-4) Qui2; H-1 MeGlc4 and H-3 Glc3; H-1 Glc5 and H-4 Xyl1; H-1 MeGlc6 and H-3 (C-3) Glc5 were observed.

The signals of C-6 MeGlc4 and C-6 Glc5 in the ^13^C NMR spectrum of **3** were observed at δ_C_ 67.0 and δ_C_ 67.1, correspondingly, due to α-shifting effects of the sulfate groups at these positions. Thus, the hexasaccharide disulfated chain of kuriloside G (**3**) was first found in the sea cucumber glycosides. The NMR spectra of the aglycone part of **3** coincided with that of kuriloside A_3_ (**1**), indicating the identity of these aglycones (Appendix A).

The (−)ESI-MS/MS of **3** demonstrated the fragmentation of [M_2Na_ − Na]^−^ ion at *m/z* 1509.5. The peaks of fragment ions were observed at *m/z* 1389.6 [M_2Na_ – Na − NaHSO_4_]^−^, 1333.5 [M_2Na_ – Na − C_7_H_12_O_5_(MeGlc)]^−^, 1231.5 [M_2Na_ – Na − C_7_H_11_O_8_SNa(MeGlcSO_3_Na)]^−^, 1069.4 [M_2Na_ – Na − C_7_H_11_O_8_SNa(MeGlcSO_3_Na) − C_6_H_10_O_5_(Glc)]^−^, 923.4 [M_2Na_ – Na − C_7_H_11_O_8_SNa(MeGlcSO_3_Na) − C_6_H_10_O_5_(Glc)] − C_6_H_10_O_4_(Qui)]^−^.

All these data indicate that kuriloside G (**3**) is 3β-*O*-{6-*O*-sodium sulfate-3-*O*-methyl-β-d-glucopyranosyl-(1→3)-β-d-glucopyranosyl-(1→4)-β-d-quinovopyranosyl-(1→2)-[3-*O*-methyl-β-d-glucopyranosyl-(1→3)-6-*O*-sodium sulfate*-*β-d-glucopyranosyl-(1→4)]-*β*-d-xylopyranosyl}-22,23,24,25,26,27-hexa-*nor*-16*α*-hydroxy,20-oxo-lanost-9(11)-ene.

The molecular formula of kuriloside H (**4**) was determined to be C_64_H_101_O_42_S_3_Na_3_ from the [M_3Na_ − Na]^−^ ion peak at *m/z* 1683.4701 (calc. 1683.4730), [M_3Na_ − 2Na]^2^^−^ ion peak at *m/z* 830.2425 (calc. 830.2419), and [M_3Na_ − 3Na]^3^^−^ ion peak at *m/z* 545.8332 (calc. 545.8315) in the (−)HR-ESI-MS. The presence of three-charged ions in the (−)HR-ESI-MS of kuriloside H (**4**) was indicative for the trisulfated glycoside.

The ^1^H and ^13^C NMR spectra corresponding to the carbohydrate chain of kuriloside H (**4**) (Table 3) demonstrated six signals of anomeric protons at δ_H_ 4.63–5.21 (d, *J* = 7.1–8.6 Hz) and the signals of anomeric carbons at δ_C_ 102.8–104.7 deduced by the HSQC spectrum, indicative of hexasaccharide moiety with *β*-glycosidic bonds. The signals of each sugar residue were assigned by the analysis of the ^1^H,^1^H-COSY, 1D TOCSY, ROESY, and HSQC spectra, enabling the identification of monosaccharide units in the chain of **4** as one xylose (Xyl1), one quinovose (Qui2), three glucoses (Glc3, Glc4 and Glc5), and one 3-*O*-methylglucose (MeGlc6). Therefore, the monosaccharide composition of **4** was the same as in kuriloside D_1_ (**2**).

However, in the ^13^C NMR spectrum of **4** three signals at δ_C_ 67.6 (C-6 Glc3), 67.4 (C-6 Glc5), and 67.0 (C-6 MeGlc6), characteristic for sulfated by C-6 hexose units, were observed instead of one signal at δ_C_ 67.0 (C-6 Glc5) in the spectrum of **2**. The signal of the OMe-group observed at δ_C_ 60.4 indicated one terminal monosaccharide residue was methylated. Actually, the protons of the OMe-group (δ_H_ 3.75, s) correlated in the HMBC spectrum with C-3 MeGlc6 (δ_C_ 86.1), which was, in turn, attached to C-3 Glc5 (ROE-correlation H-1 MeGlc6 (δ_H_ 5.13 (d, *J* = 7.4 Hz)/H-3 Glc5 (δ_H_ 4.13 (t, *J* = 8.6 Hz)). At the same time, the fourth (another terminal) monosaccharide unit was glucose (the signal of C-3 Glc4 was shielded to δ_C_ 77.7 due to the absence of *O*-methylation). The positions of all interglycosidic linkages were elucidated based on the ROESY and HMBC correlations (Table 3).

Hence, kuriloside H (**4**) has a hexasaccharide chain with a non-methylated terminal Glc4 residue and three sulfate groups. This carbohydrate chain is first found in the glycosides of the sea cucumbers and kuriloside H (**4**) is the most polar glycoside discovered so far as well as two tetrasulfated pentaosides isolated from *Psolus fabricii* [20].

The analysis of the ^13^C NMR spectrum of the aglycone part of **4** demonstrated its identity to the aglycone of kurilosides A_1_ and C_1_, isolated earlier [19]. Therefore, kuriloside H (**4**) contains a 22,23,24,25,26,27-hexa-*nor*-lanostane aglycone with 9(11)-double bond and acetoxy-groups at C-16 and C-20. *β*-orientation of the acetoxy group at C-16 and (20*S*)-configuration were established on the base of coincidence of the coupling constants (*J*_16/17_ = 7.7 Hz and *J*_17/20_ = 10.6 Hz), observed in the ^1^H NMR spectra of **4** and kuriloside A_1_, and confirmed by the ROE-correlation H-16/H-32 in the spectrum of **4** (Appendix A).

The (−)ESI-MS/MS of kuriloside H (**4**) demonstrated the fragmentation of the [M_3Na_ − Na]^−^ ion at *m/z* 1683.5. The peaks of fragment ions were observed at *m/z* 1503.5 [M_3Na_ – Na − CH_3_COOH − NaHSO_4_]^−^, 1443.5 [M_3Na_ – Na − 2CH_3_COOH − NaHSO_4_]^−^, 1281.4 [M_3Na_ – Na − 2CH_3_COOH − NaHSO_4_ − C_6_H_10_O_5_(Glc)]^−^, 1165.4 [M_3Na_ – Na − 2CH_3_COOH − NaHSO_4_ − C_7_H_11_O_8_SNa(MeGlcOSO_3_)]^−^, and 1003.4 [M_3Na_ – Na − 2CH_3_COOH − NaHSO_4_ − C_7_H_11_O_8_SNa(MeGlcOSO_3_) − C_6_H_10_O_5_(Glc)]^−^, corroborating its carbohydrate chain structure.

All these data indicate that kuriloside H (**4**) is 3β-*O*-{β-d-glucopyranosyl-(1→3)-6-*O*-sodium sulfate-β-d-glucopyranosyl-(1→4)-β-d-quinovopyranosyl-(1→2)-[6-*O*-sodium sulfate-3-*O*-methyl-β-d-glucopyranosyl-(1→3)-6-*O*-sodium sulfate*-*β-d-glucopyranosyl-(1→4)]-β-d-xylopyranosyl}-22,23,24,25,26,27-hexa-*nor*-16β,(20*S*)-diacetoxy-lanost-9(11)-ene.

The molecular formula of kuriloside I (**5**) was determined to be C_54_H_87_O_35_S_3_Na_3_ from the [M_3Na_ − Na]^−^ ion peak at *m/z* 1437.3952 (calc. 1437.3991), [M_3Na_ − 2Na]^2−^ ion peak at *m/z* 707.2049 (calc. 707.2049), and [M_3Na_ − 3Na]^3−^ ion peak at *m/z* 463.8076 (calc. 463.8069) in the (−)HR-ESI-MS, indicating the presence of three sulfate groups. The ^1^H and ^13^C NMR spectra corresponding to the carbohydrate part of kuriloside I (**5**) (Table 4) demonstrated five characteristic doublets at δ_H_ 4.63–5.13 (d, *J* = 6.6–7.8 Hz) and corresponding signals of anomeric carbons at δ_C_ 102.4–104.7 deduced by the HSQC spectrum, which indicated the presence of five monosaccharide residues in the carbohydrate chain of **5**. The signals at δ_C_ 67.0, 67.6, and 67.7 indicated the presence of three sulfate groups as in the carbohydrate chain of kuriloside H (**4**). Indeed, the comparison of the ^13^C NMR spectra of kurilosides I (**5**) and H (**4**) showed that they differed by the absence in the spectrum of **5** of the signals corresponding to non-sulfated terminal glucose residue attached to C-3 Glc3 in the carbohydrate chain of **4**. The signal of C-3 Glc3 in the ^13^C NMR spectrum of **5** was observed at δ_C_ 76.9 (instead of δ_C_ 86.3 in the spectrum of **4**), demonstrating the absence of a glycosylation effect. The presence of xylose (Xyl1), quinovose (Qui2), two glucose (Glc3, Glc4), and one 3-*O*-methylglucose (MeGlc5) residue was deduced from the analysis of the ^1^H,^1^H-COSY, HSQC and 1D TOCSY spectra of **5**. The positions of interglycosidic linkages were elucidated based on the ROESY and HMBC correlations (Table 4) and indicated the presence of the branched at the C-4 Xyl1 pentasaccharide chain in **5**, with the same architecture as in the other pentaosides of *T. kurilensis*. Thus, kuriloside I (**5**) contains a new pentasaccharide branched trisulfated chain.

The analysis of the ^13^C and ^1^H NMR spectra of the aglycone part of **5** indicated the presence of 22,23,24,25,26,27-hexa-*nor*-lanostane aglycone having a 9(11)-double bond (Table 5). The signals of methine group CH-16 were observed at c δ_C_ 72.8 (C-16) and at δ_H_ 4.82 (dd, *J* = 7.1; 14.9 Hz, H-16) due to the attachment of the hydroxyl group to this position. The HMBC correlations H-15/C-16 and H-20/C-16 confirmed this. The signals of C-20 and H-20 were shielded to δ_C_ 66.5 and δ_H_ 4.38 (dd, *J* = 6.0; 9.5 Hz), correspondingly, when compared with the same signals in the spectra of kuriloside H (**4**) (δ_C-20_ 69.4, δ_H-20_ 5.46 (dd, *J* = 6.1; 10.6 Hz)), containing (20S)-acetoxy-group. Hence, it was supposed that the attachment of the hydroxyl group to C-20 was in the aglycone of kuriloside I (**5**) instead of the acetoxy group in the aglycone of kuriloside H (**4**). 

The ROE-correlations H-16/H-17 and H-16/H-32 indicated a 16*β*-OH orientation in the aglycone of kuriloside I (**5**). (20*S*)-configuration in **5** was determined on the base of the closeness of the coupling constant *J*_20/17_ = 9.5 Hz to those in the spectra of kurilosides A_1_, C_1_ [19], and H (**4**) and corroborated by the observed ROE-correlations H-17/H-21, H-20/H-18 and biogenetic background. Hence, kuriloside I (**5**) has an aglycone with a 16*β*,(20*S*)-dihydroxy-fragment that is unique in marine glycosides.

The (−)ESI-MS/MS of kuriloside I (**5**) demonstrated the fragmentation of the [M_3Na_ − Na]^−^ ion at *m/z* 1437.5. The peaks of fragment ions were observed at *m/z* 1317.4 [M_3Na_ – Na − NaHSO_4_]^−^, 1197.4 [M_3Na_ – Na − 2NaHSO_4_]^−^, 1173.4 [M_3Na_ − Na − C_6_H_9_O_8_SNa(GlcOSO_3_)]^−^, 1039.4 [M_3Na_ – Na − NaHSO_4_ − C_7_H_11_O_8_SNa(MeGlcOSO_3_)]^−^, 1027.3 [M_3Na_ – Na − C_6_H_9_O_8_SNa(GlcOSO_3_) − C_6_H_10_O_4_(Qui)]^−^, 907.3 [M_3Na_ – Na − NaHSO_4_ − C_6_H_9_O_8_SNa(GlcOSO_3_) − C_6_H_10_O_4_(Qui)]^−^, 895.4 [M_3Na_ – Na − C_6_H_9_O_8_SNa(GlcOSO_3_) − C_7_H_11_O_8_SNa(MeGlcOSO_3_)]^−^, 667.4 [M_3Na_ – Na − C_24_H_39_O_2_(Agl) − C_6_H_9_O_8_SNa(GlcOSO_3_) − C_6_H_10_O_4_(Qui) − H]^−^, 519.0 [M_3Na_ – Na − C_24_H_39_O_3_(Agl) − C_6_H_9_O_8_SNa(GlcOSO_3_) − C_6_H_10_O_4_(Qui) − C_5_H_8_O_4_(Xyl) − H]^−^, and 417.1 [M_3Na_ – Na − C_24_H_39_O_3_(Agl) − C_6_H_9_O_8_SNa(GlcOSO_3_) − C_6_H_10_O_4_(Qui) − C_5_H_8_O_4_(Xyl) − NaHSO_3_]^−^, corroborating the structure of the glycoside.

All these data indicate that kuriloside I (**5**) is 3β-*O*-{6-*O*-sodium sulfate-β-d-glucopyranosyl-(1→4)-β-d-quinovopyranosyl-(1→2)-[6-*O*-sodium sulfate-3-*O*-methyl-β-d-glucopyranosyl-(1→3)-6-*O*-sodium sulfate-β-d-glucopyranosyl-(1→4)]-β-d-xylopyranosyl}-22,23,24,25,26,27-hexa-nor-16β,(20*S*)-dihydroxy-lanost-9(11)-ene.

The molecular formula of kuriloside I_1_ (**6**) was determined to be C_58_H_91_O_37_S_3_Na_3_ from the [M_3Na_ − 2Na]^2−^ ion peak at *m/z* 749.2148 (calc. 747.2155) and [M_3Na_ − 3Na]^3−^ ion peak at *m/z* 491.8146 (calc. 491.8139) in the (−)HR-ESI-MS. Kuriloside I_1_ (**6**) as well as kuriloside I (**5**) belong to one group because they have identical trisulfated pentasaccharide chains and, therefore, parts of the ^1^H and ^13^C NMR spectra corresponding to the carbohydrate chains are coincident (Table 4). 22,23,24,25,26,27-hexa-*nor*-lanostane aglycone of kuriloside I_1_ (**6**) is identical to that of kurilosides H (**4**), A_1_ and C_1_ [19] (Appendix A) and characterized by the presence of 16*β*,(20*S*)-diacetoxy-fragment.

The (−)ESI-MS/MS of **6** demonstrated the fragmentation of the [M_3Na_ − Na]^−^ ion at *m/z* 1521.4 and [M_3Na_ − 2Na]^2−^ ion at *m/z* 749.2. The peaks of fragment ions were observed at *m/z*: 1281.4 [M_3Na_ – Na − 2CH_3_COOH − NaHSO_4_]^−^, 1197.4 [M_3Na_ – Na − CH_3_COOH − C_6_H_9_O_8_SNa(GlcOSO_3_)]^−^, 1137.4 [M_3Na_ – Na − 2CH_3_COOH − C_6_H_9_O_8_SNa(GlcOSO_3_)]^−^, 859.4 [M_3Na_ – Na − 2CH_3_COOH − C_6_H_9_O_8_SNa(GlcOSO_3_) − C_7_H_11_O_8_SNa(MeGlcOSO_3_)]^−^, 719.2 [M_3Na_ − 2Na − CH_3_COOH]^2−^, 629.2 [M_3Na_ − 2Na − NaHSO_4_]^2−^, and 557.2 [M_3Na_ − 2Na − 2CH_3_COOH − C_6_H_9_O_8_SNa(GlcOSO_3_)]^2−^, which confirmed its structure, established by the NMR data.

All these data indicate that kuriloside I_1_ (**6**) is 3β-*O*-{6-*O*-sodium sulfate-β-d-glucopyranosyl-(1→4)-*β*-d-quinovopyranosyl-(1→2)-[6-*O*-sodium sulfate-3-*O*-methyl-β-d-glucopyranosyl-(1→3)-6-*O*-sodium sulfate-β-d-glucopyranosyl-(1→4)]-β-d-xylopyranosyl}-22,23,24,25,26,27-hexa-*nor*-16β,(20*S*)-diacetoxy-lanost-9(11)-ene.

The molecular formula of kuriloside J (**7**) was determined to be C_56_H_90_O_33_S_2_Na_2_ from the [M_2Na_−Na]^−^ ion peak at *m/z* 1377.4687 (calc. 1377.4709) and [M_2Na_−2Na]^2−^ ion peak at *m/z* 677.2413 (calc. 677.2408) in the (−)HR-ESI-MS. In the ^1^H and ^13^C NMR spectra of the carbohydrate part of kuriloside J (**7**) (Table 6), five signals of anomeric protons at δ_H_ 4.65–5.12 (d, *J* = 7.2–7.9 Hz) and corresponding five signals of anomeric carbons at δ_C_ 102.0–104.7, deduced by the HSQC spectrum, were observed, which indicated the presence of a pentasaccharide chain similar to compounds **5** and **6**. Actually, the comparison of the ^13^C NMR spectra of sugar parts of kurilosides I (**5**) and J (**7**) revealed the closeness of the signals of four monosaccharide residues, except the signals of the third unit, attached to C-4 Qui2. The analysis of the signals of this residue in the ^1^H,^1^H-COSY, HSQC, 1D TOCSY, and ROESY spectra of kuriloside J (**7**) showed that it is a glucose without a sulfate group (δ_C-6 Glc3_ 61.8, δ_C-5 Glc3_ 77.7), while in the carbohydrate chain of **5**, this residue is sulfated. The other sulfate groups occupy the same positions at C-6 Glc4 (δ_C-6 Glc4_ 67.1, δ_C-5 Glc4_ 75.1) and at C-6 MeGlc5 (δ_C-6 MeGlc5_ 66.7, δ_C-5 MeGlc5_ 75.5) as in the sugar chains of kurilosides I (**5**) and I_1_ (**6**). The positions of interglycosidic linkages in the carbohydrate chain of **7**, elucidated by the ROESY and HMBC correlations (Table 6), were the same as in kurilosides of groups A [19] and I. Thus, kuriloside J (**7**) is a branched disulfated pentaoside with the sulfate groups bonding to C-6 Glc4 and C-6 MeGlc5 in the upper semi-chain.

The analysis of the ^1^H and ^13^C NMR spectra of the aglycone part of kuriloside J (**7**) (Table 7) revealed the presence of the hexa-*nor*-lanostane aglycone having a 9(11)-double bond, similar to the majority of the other glycosides of *T. kurilensis* [19]. The signals at δ_C_ 171.2 and 21.1 were characteristic for the acetoxy group, bonded to C-16, that was deduced from the characteristic δ_C_ 75.1 value of C-16 and the ROE-correlation between the signal of *O*-acetyl methyl group (δ_H_ 2.17 (s)) and H-16 (δ_H_ 5.76 (m). Actually, in the spectrum of **7**, the signal of C-16 was deshielded by 2.3 ppm due to the presence of the acetoxy-group when compared with the corresponding signal in the spectrum of kuriloside I (**5**), having a 16-hydroxy-group. The presence of hydroxyl group at C-20 was deduced from the characteristic signals at δ_C_ 64.8 (C-20) and δ_H_ 4.28 (dd, *J* = 6.4; 10.0 Hz, H-20). Hence, the hydroxyl group is attached to C-20 in the aglycones of kuriloside I (**5**) and J (**7**). The ROE-correlation H-16/H-32 indicated 16*β*-O-Ac orientation in the aglycone of kuriloside J (**7**), which was confirmed by the coupling constant *J*_16/17_ = 7.9 Hz, indicating both protons, H-16 and H-17, to be α [21]. (20*S*)-configuration in **7** was corroborated by the coupling constant *J*_17/20_ = 10.0 Hz and the ROE-correlations H-17/H-21, H-20/H-18. Hence, kuriloside J (**7**) is characterized by the new hexa-*nor*-lanostane aglycone with a 16*β*-acetoxy,(20*S*)-hydroxy-fragment.

The (−)ESI-MS/MS of kuriloside J (**7**) demonstrated the fragmentation of [M_2Na_ − Na]^−^ ion at *m/z* 1377.5. The peaks of fragment ions were observed at *m/z* 1317.4 [M_2Na_ – Na − CH_3_COOH]^−^, 1257.4 [M_2Na_ – Na − NaHSO_4_]^−^, 1197.5 [M_2Na_ – Na − CH_3_COOH − NaHSO_4_]^−^, 1155.4 [M_2Na_ – Na − CH_3_COOH − C_6_H_10_O_5_ (Glc)]^−^, 1039.4 [M_2Na_ – Na − CH_3_COOH − C_7_H_11_O_8_SNa(MeGlcOSO_3_)]^−^, 1009.4 [M_2Na_ – Na − CH_3_COOH − C_6_H_10_O_5_(Glc) − C_6_H_10_O_4_(Qui)]^−^, 889.4 [M_2Na_ − Na − NaHSO_4_ − CH_3_COOH − C_6_H_10_O_5_(Glc) − C_6_H_10_O_4_(Qui)]^−^, 667.4 [M_2Na_ – Na − C_26_H_41_O_3_(Agl) − C_6_H_10_O_5_(Glc) − C_6_H_10_O_4_(Qui) − H]^−^, 519.0 [M_2Na_ – Na − C_26_H_41_O_3_(Agl) − C_6_H_10_O_5_(Glc) − C_6_H_10_O_4_(Qui) − C_5_H_8_O_4_(Xyl) − H]^−^, 417.1 [M_2Na_ – Na − C_26_H_41_O_3_(Agl) − C_6_H_10_O_5_(Glc) − C_6_H_10_O_4_(Qui) − C_5_H_8_O_4_(Xyl) − NaHSO_3_]^−^, corroborating the structure of its aglycone and the carbohydrate chain.

All these data indicate that kuriloside J (**7**) is 3β-*O*-{β-d-glucopyranosyl-(1→4)-β-d-quinovopyranosyl-(1→2)-[6-*O*-sodium sulfate-3-*O*-methyl-β-d-glucopyranosyl-(1→3)-6-*O*-sodium sulfate*-β*-d-glucopyranosyl-(1→4)]-β-d-xylopyranosyl}-22,23,24,25,26,27-hexa-*nor*-16β-acetoxy*,(*20*S*)-hydroxy-lanost-9(11)-ene.

The molecular formula of kuriloside K (**8**) was determined to be C_54_H_88_O_32_S_2_Na_2_ from the [M_2Na_ − Na]^−^ ion peak at *m/z* 1335.4573 (calc. 1335.4603) and the [M_2Na_ − 2Na]^2−^ ion peak at *m/z* 656.2357 (calc. 656.2356) in the (−)HR-ESI-MS. In the ^1^H and ^13^C NMR spectra of the carbohydrate part of kuriloside K (**8**) (Table 8), five signals of anomeric protons at δ_H_ 4.62–5.19 (d, *J* = 6.5–8.5 Hz) and five signals of anomeric carbons at δ_C_ 102.7–104.8, deduced by the HSQC spectrum, were indicative for the pentasaccharide chain with the *β*-configuration of glycosidic bonds. The comparison of the ^13^C NMR spectra of oligosaccharide parts of trisulfated kuriloside I (**5**) and kuriloside K (**8**) revealed the coincidence of the monosaccharide residues, except for the signals of a terminal, 3-*O*-methylglucose (MeGlc5) unit. The analysis of the signals of this residue in the ^1^H,^1^H-COSY, HSQC, 1D TOCSY, and ROESY spectra of kuriloside K (**8**) showed the absence of a sulfate group (δ_C-6 MeGlc5_ 61.6, δ_C-5 MeGlc5_ 77.5), in contrast with the carbohydrate chain of **5** (δ_C-6 MeGlc5_ 67.0, δ_C-5 MeGlc5_ 75.4). The positions of interglycosidic linkages in the carbohydrate chain of **8**, deduced by the ROESY and HMBC correlations (Table 8), showed that kuriloside K (**8**) has branching at C-4 Xyl1 in the disulfated pentasaccharide chain with the sulfate groups at C-6 Glc3 and C-6 Glc4.

The NMR spectra as well as the ROE-correlations of the aglycone part of kuriloside K (**8**) were coincident to that of kuriloside I (**5**), indicating the presence of a 22,23,24,25,26,27-hexa-*nor*-lanostane aglycone with 16β,(20*S*)-dihydroxy-fragment (Table 5).

The (−)ESI-MS/MS of **8** demonstrated the fragmentation of the [M_2Na_ − Na]^−^ ion at *m/z* 1335.4 resulted in the fragment ions observed at *m/z*: 1215.4 [M_2Na_ – Na − NaHSO_4_]^−^, 1159.4 [M_2Na_ – Na − C_7_H_12_O_5_(MeGlc)]^−^, 1071.4 [M_2Na_ – Na − C_6_H_9_O_8_SNa(GlcOSO_3_)]^−^, 925.4 [M_2Na_ – Na − C_6_H_9_O_8_SNa(GlcOSO_3_) − C_6_H_10_O_4_(Qui)]^−^, 895.4 [M_2Na_ – Na − C_6_H_9_O_8_SNa(GlcOSO_3_) − C_7_H_12_O_5_(MeGlc)]^−^, 713.3 [M_2Na_ – Na − C_24_H_39_O_2_(Agl) − C_6_H_9_O_8_SNa(GlcOSO_3_) − H]^−^, 417.1 [M_2Na_ – Na − C_24_H_39_O_3_(Agl) − C_6_H_9_O_8_SNa(GlcOSO_3_) − C_6_H_10_O_4_(Qui) − C_5_H_8_O_4_(Xyl) − H]^−^, 241.0 [M_2Na_ – Na − C_24_H_39_O_3_(Agl) − C_6_H_9_O_8_SNa(GlcOSO_3_) − C_6_H_10_O_4_(Qui) − C_5_H_8_O_4_(Xyl) − C_7_H_12_O_5_(MeGlc) − H]^−^, which confirmed the chemical structure established by the NMR data.

All these data indicate that kuriloside K (**8**) is 3β-*O*-{6-*O*-sodium sulfate-β-d-glucopyranosyl-(1→4)-β-d-quinovopyranosyl-(1→2)-[3-*O*-methyl-β-d-glucopyranosyl-(1→3)-6-*O*-sodium sulfate*-*β-d-glucopyranosyl-(1→4)]-β-d-xylopyranosyl}-22,23,24,25,26,27-hexa-*nor*-16*β,(*20*S*)-dihydroxy-lanost-9(11)-ene.

The molecular formula of kuriloside K_1_ (**9**) was determined to be C_56_H_90_O_33_S_2_Na_2_ from the [M_2Na_ − Na]^−^ ion peak at *m/z* 1377.4723 (calc. 1377.4709) and the [M_2Na_ − 2Na]^2−^ ion peak at *m/z* 677.2426 (calc. 677.2408) in the (−)HR-ESI-MS. The comparison of the ^1^H and ^13^C NMR spectra of the carbohydrate chains of kuriloside K_1_ (**9**) and kuriloside K (**8**) demonstrated their coincidence (Table 8) due to the presence of the same pentasaccharide, branched by C-4 Xyl1, sugar parts with the sulfate groups at C-6 Glc3 and C-6 Glc4. The analysis of the NMR spectra of the aglycone part of **9** indicated the presence of 22,23,24,25,26,27-hexa-*nor*-lanostane aglycone with 16*β*-acetoxy,(20*S*)-hydroxy-fragment (Table 7), identical to that of kuriloside J (**7**). Hence, kuriloside K_1_ (**9**) is an isomer of kuriloside J (**7**) by the position of one of the sulfate groups, that was confirmed by the presence of the ion-peaks having coincident *m/z* values in their (−)ESI-MS/MS spectra.

The (−)ESI-MS/MS of **9** demonstrated the fragmentation of [M_2Na_ − Na]^−^ ion at *m/z* 1377.5. The peaks of fragment ions were observed at *m/z* 1317.4 [M_2Na_ – Na − CH_3_COOH]^−^, 1197.5 [M_2Na_ – Na − CH_3_COOH − NaHSO_4_]^−^, 1069.5 [M_2Na_ – Na − C_6_H_10_O_5_(Glc)]^−^, 1053.4 [M_2Na_ – Na − CH_3_COOH − C_6_H_9_O_8_SNa(GlcOSO_3_)]^−^, 877.4 [M_2Na_ – Na − CH_3_COOH − C_6_H_9_O_8_SNa(GlcOSO_3_) − C_7_H_12_O_5_(MeGlc)]^−^, 731.3 [M_2Na_ – Na − CH_3_COOH − C_6_H_9_O_8_SNa(GlcOSO_3_) − C_7_H_12_O_5_(MeGlc) − C_6_H_10_O_4_(Qui)]^−^, 565.1 [M_2Na_ – Na − C_26_H_41_O_3_(Agl) − C_6_H_9_O_8_SNa(GlcOSO_3_) − C_6_H_10_O_4_(Qui) − H]^−^, 417.1 [M_2Na_ – Na − C_26_H_41_O_4_(Agl) − C_6_H_9_O_8_SNa(GlcOSO_3_) − C_6_H_10_O_4_(Qui) − C_5_H_8_O_4_(Xyl)]^−^.

All these data indicate that kuriloside K_1_ (**9**) is 3β-*O*-{6-*O*-sodium sulfate-β-d-glucopyranosyl-(1→4)-β-d-quinovopyranosyl-(1→2)-[3-*O*-methyl-β-d-glucopyranosyl-(1→3)-6-*O*-sodium sulfate-β-d-glucopyranosyl-(1→4)]-β-d-xylopyranosyl}-22,23,24,25,26,27-hexa-*nor*-16β-acetoxy*,*(20*S*)-hydroxy-lanost-9(11)-ene.

When the studies on the glycosides of *T. kurilensis* were started [22], the complexity of glycosidic mixture became obvious. Therefore, the part of the glycosidic sum was subjected to solvolytic desulfation to facilitate the chromatographic separation and isolation of the glycosides. However, the obtained fraction of desulfated glycosides was separated only recently as part of the effort to discover some minor glycosides possessing interesting structural peculiarities. As a result, the compounds **10** and **11** were isolated (Figure 2). Their structures were elucidated by thorough analysis of 1D and 2D NMR spectra, similar to the natural compounds **1**–**9** and confirmed by the HR-ESI-MS.

The molecular formula of DS-kuriloside L (**10**) was determined to be C_41_H_64_O_15_ from the [M − H]^−^ ion peak at *m/z* 795.4169 (calc. 795.4172) in the (−)HR-ESI-MS. Compound **10** has a trisaccharide sugar chain (for NMR data see Appendix A, for original spectra see Appendix A) and a hexa-*nor*-lanostane-type aglycone identical to that of kuriloside A_2_ [19].

The molecular formula of DS-kuriloside M (**11**) was determined to be C_54_H_88_O_26_ from the [M − H]^−^ ion peak at *m/z* 1151.5469 (calc. 1151.5491) in the (−)HR-ESI-MS. DS-kuriloside M (**11**), characterized by the 7(8)-double bond in the hexa-*nor*-lanostane nucleus and pentasaccharide chain, differed from the chains of kurilosides of the groups A, I, J, and K by the absence of sulfate groups (see Appendix A for the NMR data, Appendix A for the original spectra). Noticeably, all of the isolated kurilosides, with the exception of **11**, contained a 9(11)-double bond in the polycyclic systems.

### 2.2. Bioactivity of the Glycosides

Cytotoxic activities of compounds **1**–**9** against mouse neuroblastoma Neuro 2a, normal epithelial JB-6 cells, and erythrocytes were studied (Table 9). Known earlier cladoloside C was used as a positive control because it demonstrated a strong hemolytic effect [23]. Erythrocytes are an appropriate model for the studying of structure–activity relationships of the glycosides, since, despite many of them demonstrate hemolytic activity, the effect strongly depends on the structure of the compound. Normal epithelial JB-6 cells were used to search the compounds, not cytotoxic against this cell line, but having selective activity against other cells. Triterpene glycosides of sea cucumbers are known modulators of P2X receptors of immunocompetent cells when acting in nanomolar concentrations [24]. Neuroblastoma Neuro 2a cells are convenient model for the study of agonists/antagonists of P2X receptors—the targets in the treatment of selected nervous system diseases. Therefore, the activators, modulators, and blockers of purinergic receptors are of great interest [4] and the compounds demonstrating high cytotoxicity against Neuro 2a cells could be more deeply studied with the models of neurodegenerative diseases.

Kuriloside H (**4**), having a hexasaccharide trisulfated chain and the aglycone with acetoxy-groups at C(16) and C(20), was the most active compound in the series, demonstrating strong cytotoxicity against erythrocytes and JB-6 cells and a moderate effect against Neuro 2a cells. Kuriloside I_1_ (**6**), differing from **4** by the lack of a terminal glucose residue in the bottom semi-chain, was slightly less active. The effect of this glycoside is obviously explained by the presence of the acetoxy-group at C(20) in their aglycones, which compensates for the absence of a side chain, essential for the demonstration of the membranolytic action of the glycosides. Kurilosides J (**7**) and K_1_ (**9**), differing by the position of the second sulfate group attached to C(6) of different terminal monosaccharide residues, but having the same aglycones with 16*β*-acetoxy-group, were moderately cytotoxic against erythrocytes and JB-6 cells and had no any effect against Neuro 2a cells. However, the presence of the hydroxyl group in this position causes the loss of activity, so, the rest of compounds **1**−**3**, **5**, and **8** were not cytotoxic.

### 2.3. Biosynthetic Pathways of the Glycosides

The analysis of the structural peculiarities of the aglycones and carbohydrate chains of all the glycosides (kurilosides) found in the sea cucumber *T. kurilensis* allowed us to construct the metabolic network based on their biogenetic relationships. As a result, some biosynthetic pathways are taking shape (Figure 3).

Since the triterpene glycosides of sea cucumbers are the products of a mosaic type of biosynthesis [17], the carbohydrate chains and the aglycones are biosynthesized independently of each other. The main biosynthetic transformations of sugar parts of kurilosides are glycosylation and several rounds of sulfation that can be shifted in time relatively to each other (Figure 3). This has led to the formation of the set of compounds having 11 different oligosaccharide fragments. Meanwhile, there are some missing links (biosynthetic intermediates) in these biogenetic rows: biosides consisted of the glucose bonded to the xylose by *β*-(1→4)-glycosidic linkage, then triosides and tetraosides having glucose bonded to C(2) Xyl1—the precursors on kuriloside E, two types of disulfated hexaosides with a non-methylated terminal Glc4 unit that should biosynthetically appear between the carbohydrate chains of kurilosides of groups D and H; J and H; K and H, which have not so far been isolated. DS-kuriloside L (**10**) with a trisaccharide sugar chain is perfectly fit into the network as one of the initial stages of biosynthesis, illustrating the stepwise glycosylation of the synthesized chain. The structure of its sugar chain as well as the chain of kuriloside C_1_ [19] suggests the glycosylation of C(4) Xyl1 and initialization of the growth of the upper semi-chain precedes the glycosylation of C(2) Xyl1. There are some branchpoints of the biosynthetic pathways where the processes of sulfation and glycosylation or sulfation and methylation are alternative/concurrent. The final product of such transformations is the trisulfated hexaoside kuriloside H (**4**), the most biologically active compound in the series (Table 9), which can be formed by different pathways, and is a characteristic feature of a mosaic type of biosynthesis. However, this glycoside is minor (0.9 mg) in the glycosidic sum of *T. kurilensis*, while the main compounds are kurilosides of group A (~150 mg), and these carbohydrate chains can be considered as the most actively metabolized and resulted in the formation of at least three different types of sugar chains (kurilosides of the groups D, J, and K). Thus, their formation is a mainstream of the biosynthesis of carbohydrate chains of the glycosides of *T. kurilensis*.

As for the directions of biosynthesis of the aglycone parts of kurilosides (Figure 4), the scheme presented earlier [19] was complemented by some structures found recently, representing intermediate biosynthetic stages. DS-kuriloside M (**11**) is the only glycoside from *T. kurilensis* characterized by the 7(8)-double bond in the lanostane nucleus, when all the other kurilosides contain a 9(11)-double bond in the polycyclic systems. This finding indicates the existence of two oxidosqualene cyclases (OSCs)—enzymes converted 2,3-oxidosqualene into different triterpene alcohols giving rise various skeletons of the aglycones—in this species of sea cucumbers. These data are in good agreement with the results of the investigations of the genes coding OSCs in the other species of the sea cucumbers—*Eupentacta fraudatrix* [25], *Stichopus horrens* [26], and *Apostichopus japonicus* [27], demonstrating that even when the glycosides preferably contain the aglycones with one certain position of intra-nucleus double bond (Δ7(8)-aglycones in *E. fraudatrix* [13,18] and *S. horrens* [28,29], and Δ9(11)-aglycones in *A. japonicus* [30,31]), the genes of at least two OSCs, producing aglycone precursors with different double bond positions, are expressed, albeit with different efficiency.

The constituent hexa-*nor*-lanostane aglycones of kurilosides are biosynthesized via the oxidative cleavage of the side chain from the precursors having normal side chains (for example, kurilosides D [19] and D_1_ (**2**)) and oxygen-containing substituents at C-20 and C-22 (Figure 4). As result, the aglycone of kuriloside E [19] was formed. The subsequent biosynthetic transformations of the aglycones can occur in two directions. The first one started from the reduction of the C-20-oxo-group to the hydroxy-group, followed by the oxidation of C-16 to the hydroxy-group with the formation of the aglycones of kurilosides I (**5**) and K (**8**). It is important that the latter reaction is carried out by the cytochrome P450 monooxygenase selectively bonding to the *β*-hydroxy-group to C-16 in the derivatives containing the hydroxy-group at C-20. The next steps lead to the acetylation of hydroxyl group at C-16 (as in the aglycones of kurilosides J (**7**) and K_1_ (**9**)) followed by the acetylation of the hydroxyl group at C-20 (the aglycones of kurilosides A_1_, C_1_, H (**4**), and I_1_ (**6**) correspond to this conversion). Obviously, the oxidation of C-16 precedes the acetylation of C-20 since no aglycones with a 16-hydroxy,20-acetoxy-fragment have been found. 

The second direction of the aglycone biosynthesis occurs through the introduction of the α-hydroxyl group to C-16, resulting in the formation of aglycone of kurilosides A_3_ (**1**), G (**3**), and F [19]. Moreover, the transformation leading to hexa-*nor*-lanostane aglycones having a 16α-hydroxy,20-oxo-fragment is the same in the biosynthetic precursors with 7(8)- and 9(11)-double bonds, which is confirmed by the aglycone structure of **11**. Subsequent acetylation of the 16*α-*OH-group leads to the aglycone of kuriloside A, while intramolecular dehydration to the aglycone of kuriloside A_2_ and DS-kuriloside L (**10**). Therefore, an *α*-hydroxy-group was selectively introduced to C-16 of the 20-oxo-lanostane precursors.

## 3. Materials and Methods

### 3.1. General Experimental Procedures

Specific rotation, Perkin-Elmer 343 Polarimeter (Perkin-Elmer, Waltham, MA, USA); NMR, Bruker Avance III 700 Bruker FT-NMR (Bruker BioSpin GmbH, Rheinstetten, Germany) (700.00/176.03 MHz) (^1^H/^13^C) spectrometer; ESI MS (positive and negative ion modes), Agilent 6510 Q-TOF apparatus (Agilent Technology, Santa Clara, CA, USA), sample concentration 0.01 mg/mL; HPLC, Agilent 1260 Infinity II with a differential refractometer (Agilent Technology, Santa Clara, CA, USA); column Phenomenex Synergi Fusion RP (10 × 250 mm, 5 μm) (Phenomenex, Torrance, CA, USA).

### 3.2. Animals and Cells

Specimens of the sea cucumber *Thyonidium* (*=Duasmodactyla*) *kurilensis* (Levin) (family Cucumariidae; order Dendrochirotida) were collected in August 1990 using an industrial rake-type dredge in the waters of Onekotan Island (Kurile Islands, the Sea of Okhotsk) at a depth of 100 m by the medium fishing refrigerator trawler “Breeze” with a rear scheme of trawling during scallop harvesting. The sea cucumbers were identified by Prof. V.S. Levin; voucher specimens are preserved at the A.V. Zhirmunsky National Scientific Center of Marine Biology, Vladivostok, Russia.

CD-1 mice, weighing 18–20 g, were purchased from RAMS ‘Stolbovaya’ nursery (Stolbovaya, Moscow District, Russia) and kept at the animal facility in standard conditions. All experiments were performed following the protocol for animal study approved by the Ethics Committee of the Pacific Institute of Bioorganic Chemistry No. 0085.19.10.2020. All experiments were conducted in compliance with all of the rules and international recommendations of the European Convention for the Protection of Vertebrate Animals Used for Experimental Studies.

Mouse epithelial JB-6 cells Cl 41-5a and mouse neuroblastoma cell line Neuro 2a (ATCC ^®^ CCL-131) were purchased from ATCC (Manassas, VA, USA).

### 3.3. Extraction and Isolation

The extract of the glycosides, obtained by the standard procedure, and the initial stages of their separation were discussed in a previous paper [19]. As result of the chromatography on Si gel columns using CHCl_3_/EtOH/H_2_O (100:125:25) as the mobile phase, the fractions II–V were obtained, which were subjected to HPLC on a Phenomenex Synergi Fusion RP (10 × 250 mm) column. The separation of fraction II with MeOH/H_2_O/NH_4_OAc (1 M water solution) (63/35/2) as the mobile phase resulted in the isolation of individual kuriloside A_3_ (**1**) (79.2 mg). HPLC of fraction III with MeOH/H_2_O/NH_4_OAc (1 M water solution) (60/38/2) as the mobile phase gave 3.1 mg of kuriloside K_1_ (**9**) and 0.9 mg of kuriloside K (**8**). Fraction IV was the result of the HPLC using MeOH/H_2_O/NH_4_OAc (1 M water solution) (68/31/1) as the mobile phase was separated to the subfractions 1–7. Further rechromatography of subfraction 7 with MeOH/H_2_O/NH_4_OAc (1 M water solution) (63/34/3) followed by (60/37/3) as the mobile phases gave 3.1 mg of kuriloside I_1_ (**6**). The use of the ratio of MeOH/H_2_O/NH_4_OAc (1 M water solution) (62/35/3) for subfraction 4 gave 2.3 mg of kuriloside J (**7**) and the ratio (58/39/3) for subfraction 3 gave 7 mg of kuriloside D_1_ (**2**). For the HPLC of the most polar fraction V, obtained after Si gel chromatography, the ratio of the same solvents (60/39/1) was applied, which led to the isolation of 10 subfractions. Some of them were minor, thus only the main ones were submitted for further separation. For subfraction 10, the ratio (64/34/2) was applied to give 0.9 mg of kuriloside H (**4**). The ratio (54/43/3) used for HPLC of subfraction 4 gave 1.9 mg of kuriloside G (**3**) and 2.3 mg of kuriloside I (**5**). 

The fraction of desulfated derivatives obtained earlier by the standard methodology (~350 mg) was submitted to column chromatography on Si gel using CHCl_3_/EtOH/H_2_O (100:50:4) and CHCl_3_/MeOH/H_2_O (250:75:3) as mobile phases to give subfractions DS-1−DS-8, which were subsequently subjected to HPLC on the same column as compounds **1**–**9**. Individual DS-kuriloside M (**11**) (3.8 mg) was isolated as a result of separating the subfraction DS-6 with 66% MeOH as the mobile phase which gave several fractions, followed by the HPLC of one of them with 32% CH_3_CN as the mobile phase. HPLC of subfraction DS-2 with 50% CH_3_CN as the mobile phase, followed by 46% CH_3_CN as the mobile phase, gave 4.0 mg of DS-kuriloside L (**10**).

#### 3.3.1. Kuriloside A_3_ (**1**)

Colorless powder; ${\left[\alpha \right]}_{D}^{20}$ −1° (*c* 0.1, 50% MeOH). NMR: See Appendix A, Appendix A. (−)HR-ESI-MS *m/z*: 1231.5063 (calc. 1231.5059) [M_Na_ − Na]^−^; (−)ESI-MS/MS *m/z*: 1069.5 [M_Na_ – Na − C_6_H_10_O_5_(Glc)]^−^, 1055.4 [M_Na_ − Na–C_7_H_12_O_5_(MeGlc)]^−^, 923.4 [M_Na_ − Na–C_6_H_10_O_5_(Glc) − C_6_H_10_O_4_(Qui)]^−^, 747.3 [M_Na_ − Na–C_6_H_10_O_5_(Glc) − C_6_H_10_O_4_(Qui) − C_7_H_12_O_5_(MeGlc)]^−^, 695.1 [M_Na_ – Na − C_24_H_37_O_3_(Agl) − C_6_H_10_O_5_(Glc) − H]^−^, 565.1 [M_Na_ − Na–C_24_H_37_O_2_(Agl) − C_6_H_10_O_5_(Glc) − C_6_H_10_O_4_(Qui) − H]^−^, 549.1 [M_Na_ − Na–C_24_H_37_O_3_(Agl) − C_6_H_10_O_5_(Glc) − C_6_H_10_O_4_(Qui) − H]^−^, 417.1 [M_Na_ − Na–C_24_H_37_O_3_(Agl) − C_6_H_10_O_5_(Glc) − C_6_H_10_O_4_(Qui) − C_5_H_8_O_4_(Xyl) − H]^−^, 241.0 [M_Na_ − Na–C_24_H_37_O_3_(Agl) − C_6_H_10_O_5_(Glc) − C_6_H_10_O_4_(Qui) − C_5_H_8_O_4_(Xyl) – C_7_H_12_O_5_(MeGlc) − H]^−^.

#### 3.3.2. Kuriloside D_1_ (**2**)

Colorless powder; ${\left[\alpha \right]}_{D}^{20}$ –39° (*c* 0.1, 50% MeOH). NMR: See Table 1 and Appendix A, Appendix A. (−)HR-ESI-MS *m/z*: 1507.6291 (calc. 1507.6268) [M_Na_ − Na]^−^; (−)ESI-MS/MS *m/z*: 1349.5 [M_Na_ – Na − C_8_H_15_O_3_ + H]^−^, 1187.5 [M_Na_ – Na − C_8_H_15_O_3_ −C_6_H_10_O_5_(Glc) + H]^−^, 1025.4 [M_Na_ – Na − C_8_H_15_O_3_ − C_6_H_10_O_5_(Glc) − C_6_H_10_O_5_(Glc) + H]^−^, 879.4 [M_Na_ – Na − C_8_H_15_O_3_ −C_6_H_10_O_5_(Glc) − C_6_H_10_O_5_(Glc) − C_6_H_10_O_4_(Qui) + H]^−^, 565.1 [M_Na_ – Na − C_30_H_47_O_4_(Agl) − C_6_H_10_O_5_(Glc) − C_6_H_10_O_5_(Glc) −C_6_H_10_O_4_(Qui) − H]^−^, 417.1 [M_Na_ – Na − C_30_H_47_O_5_(Agl) − C_6_H_10_O_5_(Glc) − C_6_H_10_O_5_(Glc) − C_6_H_10_O_4_(Qui) − C_5_H_8_O_4_(Xyl) − H]^−^, 241.0 [M_Na_ – Na − C_30_H_47_O_5_(Agl) − C_6_H_10_O_5_(Glc) − C_6_H_10_O_5_(Glc) − C_6_H_10_O_4_(Qui) − C_5_H_8_O_4_(Xyl) − C_7_H_12_O_5_(MeGlc) − H]^−^.

#### 3.3.3. Kuriloside G (**3**)

Colorless powder; ${\left[\alpha \right]}_{D}^{20}$ −2° (*c* 0.1, 50% MeOH). NMR: See Table 2 and Appendix A, Appendix A. (−)HR-ESI-MS *m/z*: 1509.5102 (calc. 1509.5132) [M_2Na_ − Na]^−^; 743.2624 (calc. 743.2626) [M_2Na_ − 2Na]^2−^, (−)ESI-MS/MS *m/z*: 1389.6 [M_2Na_ – Na − NaHSO_4_]^−^, 1333.5 [M_2Na_ – Na − C_7_H_12_O_5_(MeGlc)]^−^, 1231.5 [M_2Na_ – Na − C_7_H_11_O_8_SNa(MeGlcSO_3_Na)]^−^, 1069.4 [M_2Na_ – Na − C_7_H_11_O_8_SNa(MeGlcSO_3_Na) − C_6_H_10_O_5_(Glc)]^−^, 923.4 [M_2Na_ –Na − C_7_H_11_O_8_SNa(MeGlcSO_3_Na) − C_6_H_10_O_5_(Glc)] − C_6_H_10_O_4_(Qui)]^−^.

#### 3.3.4. Kuriloside H (**4**)

Colorless powder; ${\left[\alpha \right]}_{D}^{20}$ −3° (*c* 0.1, 50% MeOH). NMR: See Table 3 and Appendix A, Appendix A. (−)HR-ESI-MS *m/z*: 1683.4701 (calc. 1683.4730) [M_3Na_ − Na]^−^, 830.2425 (calc. 830.2419) [M_3Na_ − 2Na]^2^^−^, 545.8332 (calc. 545.8315) [M_3Na_ − 3Na]^3^^−^; (−)ESI-MS/MS *m/z*: 1503.5 [M_3Na_ – Na − CH_3_COOH − NaHSO_4_]^−^, 1443.5 [M_3Na_ – Na − 2CH_3_COOH − NaHSO_4_]^−^, 1281.4 [M_3Na_ – Na − 2CH_3_COOH − NaHSO_4_ − C_6_H_10_O_5_(Glc)]^−^, 1165.4 [M_3Na_ – Na − 2CH_3_COOH − NaHSO_4_ − C_7_H_11_O_8_SNa(MeGlcOSO_3_)]^−^, 1003.4 [M_3Na_ – Na − 2CH_3_COOH − NaHSO_4_ − C_7_H_11_O_8_SNa(MeGlcOSO_3_) − C_6_H_10_O_5_ (Glc)]^−^.

#### 3.3.5. Kuriloside I (**5**)

Colorless powder; ${\left[\alpha \right]}_{D}^{20}$ −9° (*c* 0.1, 50% MeOH). NMR: See Table 4 and Table 5, Appendix A. (−)HR-ESI-MS *m/z*: 1437.3952 (calc. 1437.3991) [M_3Na_ − Na]^−^, 707.2049 (calc. 707.2049) [M_3Na_ − 2Na]^2^^−^, 463.8076 (calc. 463.8069) [M_3Na_ − 3Na]^3^^−^; (−)ESI-MS/MS *m/z*: 1317.4 [M_3Na_ – Na − NaHSO_4_]^−^, 1197.4 [M_3Na_ – Na − 2NaHSO_4_]^−^, 1173.4 [M_3Na_ – Na − C_6_H_9_O_8_SNa(GlcOSO_3_)]^−^, 1039.4 [M_3Na_ – Na − NaHSO_4_ − C_7_H_11_O_8_SNa(MeGlcOSO_3_)]^−^, 1027.3 [M_3Na_ – Na − C_6_H_9_O_8_SNa(GlcOSO_3_) − C_6_H_10_O_4_(Qui)]^−^, 907.3 [M_3Na_ – Na − NaHSO_4_ − C_6_H_9_O_8_SNa(GlcOSO_3_) − C_6_H_10_O_4_(Qui)]^−^, 895.4 [M_3Na_ – Na − C_6_H_9_O_8_SNa(GlcOSO_3_) − C_7_H_11_O_8_SNa(MeGlcOSO_3_)]^−^, 667.4 [M_3Na_ – Na − C_24_H_39_O_2_(Agl) − C_6_H_9_O_8_SNa (GlcOSO_3_) − C_6_H_10_O_4_(Qui) − H]^−^, 519.0 [M_3Na_ – Na − C_24_H_39_O_3_(Agl) − C_6_H_9_O_8_SNa(GlcOSO_3_) − C_6_H_10_O_4_(Qui) − C_5_H_8_O_4_(Xyl) − H]^−^, 417.1 [M_3Na_ – Na − C_24_H_39_O_3_(Agl) − C_6_H_9_O_8_SNa(GlcOSO_3_) − C_6_H_10_O_4_(Qui) − C_5_H_8_O_4_(Xyl) − NaHSO_3_]^−^.

#### 3.3.6. Kuriloside I_1_ (**6**)

Colorless powder; ${\left[\alpha \right]}_{D}^{20}$ −5° (*c* 0.1, 50% MeOH). NMR: See Table 4 and Appendix A, Appendix A. (−)HR-ESI-MS *m/z*: 1749.2148 (calc. 747.2155) [M_3Na_ − 2Na]^2−^, 491.8146 (calc. 491.8139) [M_3Na_ − 3Na]^3−^; (−)ESI-MS/MS *m/z*: 1281.4 [M_3Na_ – Na − 2CH_3_COOH − NaHSO_4_]^−^, 1197.4 [M_3Na_ – Na − CH_3_COOH − C_6_H_9_O_8_SNa(GlcOSO_3_)]^−^, 1137.4 [M_3Na_ – Na − 2CH_3_COOH − C_6_H_9_O_8_SNa(GlcOSO_3_)]^−^, 859.4 [M_3Na_ – Na − 2CH_3_COOH − C_6_H_9_O_8_SNa(GlcOSO_3_) − C_7_H_11_O_8_SNa(MeGlcOSO_3_)]^−^, 719.2 [M_3Na_ − 2Na − CH_3_COOH]^2−^, 629.2 [M_3Na_ − 2Na − NaHSO_4_]^2−^, 557.2 [M_3Na_ − 2Na − 2CH_3_COOH − C_6_H_9_O_8_SNa(GlcOSO_3_)]^2−^.

#### 3.3.7. Kuriloside J (**7**)

Colorless powder; ${\left[\alpha \right]}_{D}^{20}$ −10° (*c* 0.1, 50% MeOH). NMR: See Table 6 and Table 7, Appendix A. (−)HR-ESI-MS *m/z*: 1377.4687 (calc. 1377.4709) [M_2Na_ − Na]^−^, 677.2413 (calc. 677.2408) [M_2Na_ − 2Na]^2−^; (−)ESI-MS/MS *m/z*: 1317.4 [M_2Na_ – Na − CH_3_COOH]^−^, 1257.4 [M_2Na_ – Na − NaHSO_4_]^−^, 1197.5 [M_2Na_ – Na − CH_3_COOH − NaHSO_4_]^−^, 1155.4 [M_2Na_ – Na − CH_3_COOH − C_6_H_10_O_5_(Glc)]^−^, 1039.4 [M_2Na_ – Na − CH_3_COOH − C_7_H_11_O_8_SNa(MeGlcOSO_3_)]^−^, 1009.4 [M_2Na_ – Na − CH_3_COOH − C_6_H_10_O_5_(Glc) − C_6_H_10_O_4_(Qui)]^−^, 889.4 [M_2Na_ – Na − NaHSO_4_ − CH_3_COOH − C_6_H_10_O_5_(Glc) − C_6_H_10_O_4_(Qui)]^−^, 667.4 [M_2Na_ – Na − C_26_H_41_O_3_(Agl) − C_6_H_10_O_5_(Glc) − C_6_H_10_O_4_(Qui) − H]^−^, 519.0 [M_2Na_ – Na − C_26_H_41_O_3_(Agl) − C_6_H_10_O_5_(Glc) − C_6_H_10_O_4_(Qui) − C_5_H_8_O_4_(Xyl) − H]^−^, 417.1 [M_2Na_ – Na − C_26_H_41_O_3_(Agl) − C_6_H_10_O_5_ (Glc) − C_6_H_10_O_4_ (Qui) − C_5_H_8_O_4_ (Xyl) − NaHSO_3_]^−^.

#### 3.3.8. Kuriloside K (**8**)

Colorless powder; ${\left[\alpha \right]}_{D}^{20}$ −7° (*c* 0.1, 50% MeOH). NMR: See Table 5 and Table 8, Appendix A. (−)HR-ESI-MS *m/z*: 1335.4573 (calc. 1335.4603) [M_2Na_ − Na]^−^, 656.2357 (calc. 656.2356) [M_2Na_ − 2Na]^2−^; (−)ESI-MS/MS *m/z*: 1215.4 [M_2Na_ – Na − NaHSO_4_]^−^, 1159.4 [M_2Na_ – Na − C_7_H_12_O_5_(MeGlc)]^−^, 1071.4 [M_2Na_ – Na − C_6_H_9_O_8_SNa(GlcOSO_3_)]^−^, 925.4 [M_2Na_ – Na − C_6_H_9_O_8_SNa (GlcOSO_3_) − C_6_H_10_O_4_ (Qui)]^−^, 895.4 [M_2Na_ – Na − C_6_H_9_O_8_SNa(GlcOSO_3_) − C_7_H_12_O_5_(MeGlc)]^−^, 713.3 [M_2Na_ – Na − C_24_H_39_O_2_(Agl) − C_6_H_9_O_8_SNa(GlcOSO_3_) − H]^−^, 417.1 [M_2Na_ – Na − C_24_H_39_O_3_(Agl) − C_6_H_9_O_8_SNa(GlcOSO_3_) − C_6_H_10_O_4_(Qui) − C_5_H_8_O_4_(Xyl) − H]^−^, 241.0 [M_Na_ – Na − C_24_H_39_O_3_(Agl) − C_6_H_9_O_8_SNa(GlcOSO_3_) − C_6_H_10_O_4_(Qui) − C_5_H_8_O_4_(Xyl) − C_7_H_12_O_5_(MeGlc) − H]^−^.

#### 3.3.9. Kuriloside K_1_ (**9**)

Colorless powder; ${\left[\alpha \right]}_{D}^{20}$ −4° (*c* 0.1, 50% MeOH). NMR: See Table 7 and Table 8, Appendix A. (−)HR-ESI-MS *m/z*: 1377.4723 (calc. 1377.4709) [M_2Na_ − Na]^−^, 677.2426 (calc. 677.2408) [M_2Na_ − 2Na]^2−^; (−)ESI-MS/MS *m/z*: 1317.4 [M_2Na_ – Na − CH_3_COOH]^−^, 1197.5 [M_2Na_ – Na − CH_3_COOH − NaHSO_4_]^−^, 1069.5 [M_2Na_ – Na − C_6_H_10_O_5_ (Glc)]^−^, 1053.4 [M_2Na_ – Na − CH_3_COOH − C_6_H_9_O_8_SNa(GlcOSO_3_)]^−^, 877.4 [M_2Na_ – Na − CH_3_COOH −C_6_H_9_O_8_SNa(GlcOSO_3_) − C_7_H_12_O_5_(MeGlc)]^−^, 731.3 [M_2Na_ – Na − CH_3_COOH − C_6_H_9_O_8_SNa(GlcOSO_3_) − C_7_H_12_O_5_(MeGlc) − C_6_H_10_O_4_(Qui)]^−^, 565.1 [M_2Na_ – Na − C_26_H_41_O_3_(Agl) − C_6_H_9_O_8_SNa(GlcOSO_3_) − C_6_H_10_O_4_(Qui) − H]^−^, 417.1 [M_2Na_ – Na − C_26_H_41_O_4_(Agl) − C_6_H_9_O_8_SNa(GlcOSO_3_) − C_6_H_10_O_4_(Qui) − C_5_H_8_O_4_(Xyl)]^−^.

### 3.4. Cytotoxic Activity (MTT Assay)

All compounds (including cladoloside C used as the positive control) were tested in concentrations from 1.5 μM to 100 μM using two-fold dilution in dH_2_O. The solutions (20 µL) of tested substances in different concentrations and cell suspension (180 µL) were added in wells of 96-well plates (1 × 10^4^ cells/well) and incubated 24 h at 37 °C and 5% CO_2_. After incubation, the medium with tested substances was replaced by 100 μL of fresh medium. Then, 10 μL of MTT (thiazoyl blue tertrazolium bromide) stock solution (5 mg/mL) was added to each well and the microplate was incubated for 4 h. After that, 100 μL of SDS-HCl solution (1 g SDS/10 mL dH_2_O/17 μL 6 N HCl) was added to each well followed by incubation for 4–18 h. The absorbance of the converted dye formazan was measured using a Multiskan FC microplate photometer (Thermo Fisher Scientific, Waltham, MA, USA) at a wavelength of 570 nm. Cytotoxic activity of the substances was calculated as the concentration that caused 50% metabolic cell activity inhibition (IC_50_). All the experiments were made in triplicate, *p* < 0.01.

### 3.5. Hemolytic Activity

Blood was taken from CD-1 mice (18–20 g). Erythrocytes were isolated from the blood of albino CD-1 mice by centrifugation with phosphate-buffered saline (pH 7.4) for 5 min at 4 °C by 450× *g* on a LABOFUGE 400R (Heraeus, Hanau, Germany) centrifuge for three times. Then, the residue of erythrocytes was resuspended in ice cold phosphate saline buffer (pH 7.4) to a final optical density of 1.5 at 700 nm, and kept on ice. For the hemolytic assay, 180 µL of erythrocyte suspension was mixed with 20 µL of test compound solution (including cladoloside C used as the positive control) in V-bottom 96-well plates. After 1 h of incubation at 37 °C, plates were exposed to centrifugation for 10 min at 900× *g* on a LMC-3000 (Biosan, Riga, Latvia) laboratory centrifuge. Then, we carefully selected 100 µL of supernatant and transferred it to new flat-plates respectively. Lysis of erythrocytes was determined by measuring the concentration of hemoglobin in the supernatant with a microplate photometer Multiskan FC (Thermo Fisher Scientific, Waltham, MA, USA), λ = 570 nm. The effective dose causing 50% hemolysis of erythrocytes (ED_50_) was calculated using the computer program SigmaPlot 10.0. All experiments were made in triplicate, *p* < 0.01.

### 3.6. Solvolytic Desulfation

A part of the glycosidic sum (350 mg) was dissolved in a mixture of pyridine/dioxane (1/1) and refluxed for 1 h. The obtained mixture was concentrated in vacuo and subsequently purified by using Si gel column chromatography (as depicted in the Section 3.3).

## 4. Conclusions

Thus, nine unknown earlier triterpene glycosides were isolated from the sea cucumber *Thyonidium (=Duasmodactyla) kurilensis* in addition to the series of kurilosides found recently [19]. Five new types of the carbohydrate chains (kurilosides of the groups G–K) were discovered. There were trisulfated penta- (kurilosides of the group I (**5**, **6**)) and hexaosides (kuriloside H (**4**)) among them. Kuriloside H (**4**) is the second example of the most polar triterpene glycosides, along with tetrasulfated pentaosides found earlier in the sea cucumber *Psolus fabricii* [20]. The structures of disulfated hexa- and pentasaccharide chains of kurilosides of the groups G (**3**), J (**7**), and K (**8**, **9**) clearly illustrate a combinatorial (mosaic) type of biosynthesis of the glycosides, namely, the positions of the sulfate group attachment. At the same time, the position of one of the sulfate groups (at C(6) Glc, attached to C(4) Xyl1) remained the same in all glycosides found in this species. Three new non-holostane aglycones lacking a lactone ring, two of them being the 22,23,24,25,26,27-hexa-*nor*-lanostane type and one having a normal side chain, were found in glycosides **1**–**9**. The majority of the aglycones of *T. kurilensis* glycosides differed from each other in the substituents at C-16 (*α*- and *β*-oriented hydroxy- or acetoxy groups, or keto-group) and C-20 (hydroxy-, acetoxy-, or keto-groups), representing the biogenetically related rows of the compounds. As mentioned in a previous paper [19], the glycosides with 16*α*-substituents were isolated from *T. kurilensis* only. The finding of 16*β*-hydroxylated aglycones is also for the first time. Such compounds can be considered as “hot metabolites”, biosynthetic intermediates or precursors of the aglycones with the 16*β*-acetoxy-group.

## Figures and Tables

**Figure 1 marinedrugs-19-00187-f001:**
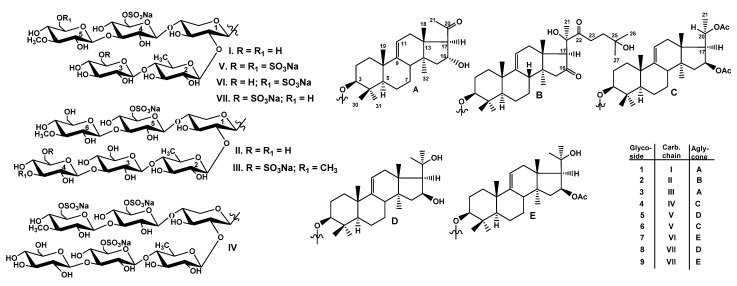
Chemical structures of glycosides isolated from *Thyonidium kurilensis:*
**1**—kuriloside A_3_; **2**—kuriloside D_1_; **3**—kuriloside G; **4**—kuriloside H; **5**—kuriloside I; **6**—kuriloside I_1_; **7**—kuriloside J, **8**—kuriloside K, **9**—kuriloside K_1_.

**Figure 2 marinedrugs-19-00187-f002:**
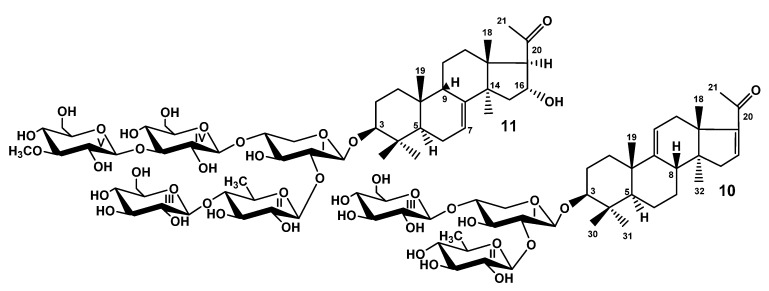
Chemical structures of desulfated glycosides isolated from *Thyonidium kurilensis:*
**10**—DS-kuriloside L; **11**—DS-kuriloside M.

**Figure 3 marinedrugs-19-00187-f003:**
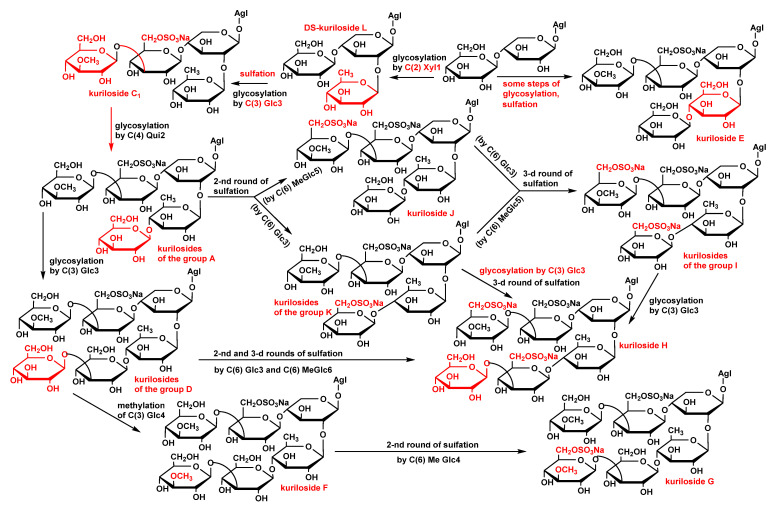
The metabolic network of the carbohydrate chains of the glycosides from *T. kurilensis.*

**Figure 4 marinedrugs-19-00187-f004:**
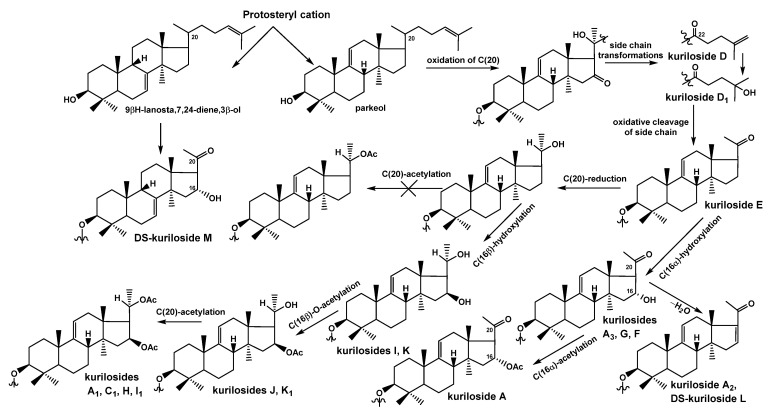
The biosynthetic pathways to aglycones of glycosides from *T. kurilensis.*

**Table 1 marinedrugs-19-00187-t001:** ^13^C and ^1^H NMR chemical shifts, HMBC, and ROESY correlations of the aglycone moiety of kuriloside D_1_ (**1**).

Position	δ_C_ mult. ^a^	δ_H_ mult. (*J* in Hz) ^b^	HMBC	ROESY
1	36.0 CH_2_	1.77 brd (12.8)		H-11, H-19
		1.39 m		H-3, H-5, H-11
2	26.9 CH_2_	2.20 m		
		1.94 brdd (11.3; 12.8)		H-19
3	88.4 CH	3.20 dd (3.8; 11.3)	C: 4, 30, 31, C: 1 Xyl1	H-1, H-5, H-31, H-1 Xyl1
4	39.7 C			
5	52.7 CH	0.90 m	C: 6, 19, 30	H-1, H-3, H-7, H-31
6	21.0 CH_2_	1.69 m		
		1.44 m		H-8, H-19
7	28.2 CH	1.49 m		
		1.28 m		H-5
8	40.2 CH	2.33 m		H-18, H-19
9	149.0 C			
10	39.4 C			
11	114.9 CH	5.36 brd (6.0)	C: 8, 10, 13	H-1
12	36.5 CH_2_	2.43 brd (16.5)		H-17, H-32
		2.20 brdd (6.0; 16.5)	C: 9, 11, 14	H-18, H-21
13	43.7 C			
14	41.9 C			
15	48.1 CH_2_	2.27 d (16.3)	C: 14, 16, 32	
		2.03 d (18.0)	C: 13, 16, 32	H-7, H-32
16	216.3 C			
17	63.8 CH	3.69 s	C: 22	H-12, H-21, H-32
18	16.9 CH_3_	1.30 s	C: 17	H-8, H-12, H-15, H-19, H-21
19	22.2 CH_3_	1.12 s	C: 1, 5, 9, 10	H-1, H-2, H-6, H-8
20	80.8 C			
21	24.7 CH_3_	1.61 s	C: 17, 20, 22	H-12, H-17, H-18, H-23
22	217.6 C			
23	32.2 CH_2_	3.67 dd (10.6; 18.2)	C: 22, 24	H-26, H-27
		3.43 dt (7.8; 18.2)	C: 22, 24	H-21, H-26, H-27
24	38.0 CH_2_	2.27 t (7.8)	C: 22, 23, 25, 26, 27	
25	69.0 C			
26	30.0 CH_3_	1.42 s	C: 24, 25, 27	H-23, H-24, H-27
27	29.5 CH_3_	1.41 s	C: 24, 25, 26	H-23, H-24, H-26
30	16.5 CH_3_	1.06 s	C: 3, 4, 5, 31	H-2, H-6, H-31
31	27.9 CH_3_	1.26 s	C: 3, 4, 5, 30	H-3, H-5, H-6, H-30, H-1 Xyl1
32	18.6 CH_3_	0.89 s	C: 8, 13, 14, 15	H-7, H-12, H-15, H-17

^a^ Recorded at 176.03 MHz in C_5_D_5_N/D_2_O (4/1). ^b^ Recorded at 700.00 MHz in C_5_D_5_N/D_2_O (4/1).

**Table 2 marinedrugs-19-00187-t002:** ^13^C and ^1^H NMR chemical shifts, HMBC, and ROESY correlations of carbohydrate moiety of kuriloside G (**3**).

Atom	δ_C_ mult.^a,b,c^	δ_H_ mult. (*J* in Hz) ^d^	HMBC	ROESY
Xyl1 (1→C-3)				
1	104.7 CH	4.65 d (7.8)	C: 3	H-3; H-3, 5 Xyl1
2	**82.2** CH	3.95 t (8.8)	C: 1 Qui2	H-1 Qui2; H-4 Xyl1
3	75.1 CH	4.15 t (8.8)	C: 4 Xyl1	H-5 Xyl1
4	**77.6** CH	4.15 m		H-1 Glc5
5	63.5 CH_2_	4.36 dd (5.4; 10.8)		
		3.61 m		H-1 Xyl1
Qui2 (1→2Xyl1)				
1	104.4 CH	5.03 d (7.8)	C: 2 Xyl1	H-2 Xyl1; H-3, 5 Qui2
2	75.2 CH	3.89 t (7.8)	C: 3 Qui2	
3	75.2 CH	3.99 t (9.3)	C: 4 Qui2	
4	**86.4** CH	3.56 t (9.3)	C: 1 Glc3	H-1 Glc3; H-2 Qui2
5	71.4 CH	3.69 dd (6.2; 9.3)		H-1 Qui2
6	17.9 CH_3_	1.64 d (6.2)	C: 4, 5 Qui2	
Glc3 (1→4Qui2)				
1	104.0 CH	4.85 d (8.1)	C: 4 Qui2	H-4 Qui2; H-3, 5 Glc3
2	73.5 CH	3.90 t (8.1)		
3	**87.5** CH	4.12 t (8.1)	C: 4 Glc3	H-1 MeGlc4; H-1, 5 Glc3
4	69.3 CH	3.84 t (8.1)	C: 5 Glc3	
5	77.5 CH	3.90 t (8.1)		
6	61.6 CH_2_	4.34 d (11.6)		
		4.03 dd (7.0; 11.6)		
MeGlc4 (1→3Glc3)				
1	104.8 CH	5.12 d (8.1)	C: 3 Glc3	H-3 Glc3; H-3, 5 MeGlc4
2	74.5 CH	3.79 t (8.1)		
3	86.3 CH	3.65 t (8.1)	C: 4 MeGlc4; OMe	H-1 MeGlc4
4	69.9 CH	3.98 t (8.1)	C: 3, 5 MeGlc4	
5	75.5 CH	4.04 t (8.1)		H-1 MeGlc4
6	*67.0* CH_2_	4.97 d (11.6)		
		4.73 dd (4.7; 11.6)		
OMe	60.5 CH_3_	3.76 s	C: 3 MeGlc4	
Glc5 (1→4Xyl1)				
1	102.1 CH	4.87 d (7.0)	C: 4 Xyl1	H-4 Xyl1; H-3, 5 Glc5
2	73.2 CH	3.84 t (8.1)	C: 1, 3 Glc5	
3	**85.9** CH	4.16 t (8.1)	C: 1 MeGlc6; C: 2, 4 Glc5	H-1 MeGlc6; H-1 Glc5
4	69.2 CH	3.91 t (9.3)	C: 5 Glc5	H-6 Glc5
5	75.5 CH	4.02 m		H-1 Glc5
6	*67.1* CH_2_	4.93 d (11.6)		
		4.69 dd (5.8; 11.6)	C: 5 Glc5	
MeGlc6 (1→3Glc5)				
1	104.4 CH	5.19 d (8.1)	C: 3 Glc5	H-3 Glc5; H-3, 5 MeGlc6
2	74.3 CH	3.84 t (8.1)		
3	86.9 CH	3.66 t (8.1)	OMe	H-1 MeGlc6
4	70.2 CH	3.89 t (8.1)	C: 5 MeGlc6	
5	77.5 CH	3.90 t (8.1)		H-1 MeGlc6
6	61.7 CH_2_	4.34 d (11.6)		
		4.06 dd (5.8; 11.6)		
OMe	60.5 CH_3_	3.80 s	C: 3 MeGlc6	

^a^ Recorded at 176.03 MHz in C_5_D_5_N/D_2_O (4/1). ^b^ Bold = interglycosidic positions. ^c^ Italic = sulfate position. ^d^ Recorded at 700.00 MHz in C_5_D_5_N/D_2_O (4/1). Multiplicity by 1D TOCSY.

**Table 3 marinedrugs-19-00187-t003:** ^13^C and ^1^H NMR chemical shifts, HMBC, and ROESY correlations of carbohydrate moiety of kuriloside H (**4**).

Atom	δ_C_ mult. ^a,b,c^	δ_H_ mult. (*J* in Hz) ^d^	HMBC	ROESY
Xyl1 (1→C-3)				
1	104.7 CH	4.63 d (8.3)	C: 3	H-3; H-3, 5 Xyl1
2	**82.6** CH	3.83 t (7.1)	C: 1 Qui2	H-1 Qui2; H-4 Xyl1
3	75.1 CH	4.05 m	C: 4 Xyl1	H-1, 5 Xyl1
4	**79.4** CH	4.04 m		H-1 Glc5
5	63.5 CH_2_	4.34 brd (10.1)	C: 3 Xyl1	
		3.59 m		H-1 Xyl1
Qui2 (1→2Xyl1)				
1	104.5 CH	4.88 d (7.1)	C: 2 Xyl1	H-2 Xyl1; H-3, 5 Qui2
2	75.4 CH	3.88 t (8.9)	C: 1, 3 Qui2	H-4 Qui2
3	74.9 CH	3.98 t (8.9)		H-1, 5 Qui2
4	**86.5** CH	3.35 t (8.9)	C: 1 Glc3	H-1 Glc3; H-2 Qui2
5	71.4 CH	3.63 dd (5.9; 8.9)		H-1, 3 Qui2
6	17.6 CH_3_	1.58 d (5.8)	C: 4, 5 Qui2	
Glc3 (1→4Qui2)				
1	104.1 CH	4.72 d (8.5)	C: 4 Qui2	H-4 Qui2; H-3, 5 Glc3
2	73.4 CH	3.83 t (8.5)		
3	**86.3** CH	4.18 t (8.5)	C: 1 Glc4; C: 2, 4 Glc3	H-1 Glc4; H-1, 5 Glc3
4	69.3 CH	3.76 t (8.5)	C: 5, 6 Glc3	
5	74.7 CH	4.10 m	C: 4 Glc3	H-1 Glc3
6	*67.6* CH_2_	4.96 d (11.0)		
		4.57 d (11.0)		
Glc4 (1→3Glc3)				
1	104.5 CH	5.21 d (8.5)	C: 3 Glc3	H-3 Glc3; H-3, 5 Glc4
2	74.7 CH	3.93 t (8.5)	C: 1, 3 Glc4	
3	77.7 CH	4.11 t (8.5)	C: 4 MeGlc4	H-1, 5 Glc4
4	71.0 CH	3.94 t (8.5)		
5	77.7 CH	3.89 m		H-1 Glc4
6	61.9 CH_2_	4.37 d (12.3)		
		4.06 dd (6.2; 12.3)		
Glc5 (1→4Xyl1)				
1	102.8 CH	4.85 d (8.6)	C: 4 Xyl1	H-4 Xyl1; H-3, 5 Glc5
2	73.2 CH	3.83 t (8.6)	C: 1, 3 Glc5	
3	**86.3** CH	4.13 t (8.6)	C: 1 MeGlc6; C: 2 Glc5	H-1 MeGlc6; H-1 Glc5
4	69.2 CH	3.76 t (8.6)		H-6 Glc5
5	77.2 CH	4.09 t (8.6)		H-1 Glc5
6	*67.4* CH_2_	4.96 d (11.1)		
		4.56 d (11.1)		H-4 Glc5
MeGlc6 (1→3Glc5)				
1	104.4 CH	5.13 d (7.4)	C: 3 Glc5	H-3 Glc5; H-3, 5 MeGlc6
2	74.3 CH	3.78 t (7.4)	C: 1 MeGlc6	H-4 MeGlc6
3	86.1 CH	3.63 t (8.6)	C: 4 MeGlc6; OMe	H-1, 5 MeGlc6; OMe
4	69.8 CH	4.00 t (8.6)	C: 3, 5 MeGlc6	
5	75.6 CH	4.00 m		H-1, 3 MeGlc6
6	*67.0* CH_2_	4.93 d (9.9)		
		4.75 dd (3.7; 11.1)		H-4 MeGlc6
OMe	60.4 CH_3_	3.75 s	C: 3 MeGlc6	

^a^ Recorded at 176.04 MHz in C_5_D_5_N/D_2_O (4/1). ^b^ Bold = interglycosidic positions. ^c^ Italic = sulfate position. ^d^ Recorded at 700.13 MHz in C_5_D_5_N/D_2_O (4/1). Multiplicity by 1D TOCSY.

**Table 4 marinedrugs-19-00187-t004:** ^13^C and ^1^H NMR chemical shifts, HMBC, and ROESY correlations of carbohydrate moiety of kurilosides I (**5**) and I_1_ (**6**).

Atom	δ_C_ mult. ^a,b,c^	δ_H_ mult. (*J* in Hz) ^d^	HMBC	ROESY
Xyl1 (1→C-3)				
1	104.7 CH	4.63 d (6.6)	C: 3	H-3; H-3, 5 Xyl1
2	**82.1** CH	3.90 t (6.6)	C: 1 Qui2	H-1 Qui2
3	75.1 CH	4.10 t (6.6)	C: 4 Xyl1	
4	**78.5** CH	4.10 t (6.6)		H-1 Glc4
5	63.4 CH_2_	4.34 dd (9.6; 11.7)	C: 3 Xyl1	
		3.59 t (11.2)		H-1 Xyl1
Qui2 (1→2Xyl1)				
1	104.4 CH	4.96 d (7.1)	C: 2 Xyl1	H-2 Xyl1; H-5 Qui2
2	75.4 CH	3.84 t (8.3)	C: 1, 3 Qui2	H-4 Qui2
3	74.9 CH	3.93 t (8.3)	C: 2, 4 Qui2	H-1, 5 Qui2
4	**87.0** CH	3.36 t (8.3)	C: 1 Glc3; C: 5, 6 Qui2	H-1 Glc3
5	71.3 CH	3.65 dd (5.9; 9.5)	C: 6 Qui2	H-1, 3 Qui2
6	17.7 CH_3_	1.59 d (5.9)	C: 4, 5 Qui2	
Glc3 (1→4Qui2)				
1	104.7 CH	4.69 d (7.8)	C: 4 Qui2	H-4 Qui2; H-5 Glc3
2	74.1 CH	3.80 t (8.6)	C: 1, 3 Glc3	
3	76.9 CH	4.11 t (8.6)	C: 2, 4 Glc3	
4	70.8 CH	3.89 t (8.6)	C: 3, 5, 6 Glc3	H-2, 6 Glc3
5	75.5 CH	4.09 t (8.6)		
6	*67.6* CH_2_	5.05 brd (9.5)		
		4.64 dd (7.8; 12.1)	C: 5 Glc3	
Glc4 (1→4Xyl1)				
1	102.4 CH	4.85 d (7.8)	C: 4 Xyl1	H-4 Xyl1; H-5 Glc4
2	73.1 CH	3.82 t (8.6)	C: 1 Glc4	
3	86.3 CH	4.12 t (8.6)	C: 1 MeGlc5; C: 4 Glc4	H-1 MeGlc5
4	69.2 CH	3.81 t (8.6)	C: 3, 5, 6 Glc4	
5	74.9 CH	4.06 t (8.6)		H-1 Glc4
6	*67.4* CH_2_	4.95 d (10.3)		
		4.61 dd (5.2; 10.3)		
MeGlc5 (1→3Glc4)				
1	104.5 CH	5.13 d (7.8)	C: 3 Glc4	H-3 Glc4; H-3, 5 MeGlc5
2	74.2 CH	3.77 t (8.6)	C: 1, 3 MeGlc5	H-4 MeGlc5
3	86.3 CH	3.62 t (8.6)	C: 2, 4 MeGlc5; OMe	H-1, 5 MeGlc5; OMe
4	69.8 CH	4.00 t (8.6)	C: 3, 5 MeGlc5	H-2, 6 MeGlc5
5	75.4 CH	4.01 t (8.6)		H-1 MeGlc5
6	*67.0* CH_2_	4.92 d (10.3)	C: 4, 5 MeGlc5	
		4.74 dd (3.5; 10.3)	C: 5 MeGlc5	
OMe	60.4 CH_3_	3.75 s	C: 3 MeGlc5	

^a^ Recorded at 176.04 MHz in C_5_D_5_N/D_2_O (4/1). ^b^ Bold = interglycosidic positions. ^c^ Italic = sulfate position. ^d^ Recorded at 700.13 MHz in C_5_D_5_N/D_2_O (4/1). Multiplicity by 1D TOCSY.

**Table 5 marinedrugs-19-00187-t005:** ^13^C and ^1^H NMR chemical shifts, HMBC, and ROESY correlations of the aglycone moiety of kurilosides I (**5**) and K (**8**).

Position	δ_C_ mult. ^a^	δ_H_ mult. (*J* in Hz) ^b^	HMBC	ROESY
1	36.2 CH_2_	1.67 m		H-11, H-30
		1.28 m		H-3, H-11
2	26.8 CH_2_	2.07 m		
		1.83 brd (11.3)		H-19, H-30
3	88.7 CH	3.09 dd (4.2; 11.3)	C: 31, C: 1 Xyl1	H-5, H-31, H1-Xyl1
4	39.6 C			
5	52.8 CH	0.75 brd (12.5)	C: 6, 7, 30	H-3, H-7, H-31
6	21.1 CH_2_	1.56 m		H-31
		1.28 dt (2.4; 12.5)		H-8, H-30
7	28.0 CH_2_	1.52 m		
		1.17 m		H-5, H-32
8	41.2 CH	2.14 m		H-18, H-19
9	148.6 C			
10	39.1 C			
11	114.3 CH	5.15 brd (6.0)	C: 10, 12, 14	H-1
12	36.4 CH_2_	1.98 brdd (3.0; 16.7)		H-32
		1.68 brd (16.7)	C: 9, 11	H-8, H-18, H-21
13	45.1 C			
14	43.2 C			
15	45.0 CH_2_	2.07 dd (7.8; 12.5)	C: 14, 17, 32	H-32
		1.70 d (6.0; 12.5)	C: 13, 14, 16, 32	
16	72.8 CH	4.82 dd (7.1; 14.9)	C: 14	H-17, H-32
17	57.2 CH	2.11 m	C: 14, 18, 20, 21	H-21, H-32
18	16.0 CH_3_	0.86 s	C: 12, 14, 15, 17	H-8, H-12, H-20, H-21
19	22.2 CH_3_	0.97 s	C: 1, 5, 9, 10	H-1, H-2, H-8, H-18
20	66.5 CH	4.38 dd (6.0; 9.5)	C: 16, 17	H-18, H-21
21	22.7 CH_3_	1.40 d (6.0)	C: 17, 20	H-12, H-17, H-18, H-20
30	16.5 CH_3_	0.96 s	C: 3, 4, 5, 31	H-2, H-6, H-31, H-6 Qui2
31	27.9 CH_3_	1.12 s	C: 3, 4, 5, 30	H-3, H-5, H-6, H-30, H-1 Xyl1
32	18.8 CH_3_	0.67 s	C: 8, 13, 14, 15	H-7, H-15, H-17

^a^ Recorded at 176.03 MHz in C_5_D_5_N/D_2_O (4/1). ^b^ Recorded at 700.00 MHz in C_5_D_5_N/D_2_O (4/1).

**Table 6 marinedrugs-19-00187-t006:** ^13^C and ^1^H NMR chemical shifts, HMBC, and ROESY correlations of carbohydrate moiety of kuriloside J (**7**).

Atom	δ_C_ mult. ^a,b,c^	δ_H_ mult. (*J* in Hz) ^d^	HMBC	ROESY
Xyl1 (1→C-3)				
1	104.4 CH	4.65 d (7.6)	C: 3	H-3; H-3, 5 Xyl1
2	**82.1** CH	3.97 t (7.6)	C: 1 Qui2	H-1 Qui2
3	75.1 CH	4.15 t (7.3)	C: 4 Xyl1	
4	**77.5** CH	4.15 t (7.3)	C: 3 Xyl1	H-1 Glc4
5	63.4 CH_2_	4.36 brd (9.3)		
		3.60 m		H-1, 3 Xyl1
Qui2 (1→2Xyl1)				
1	104.5 CH	5.07 d (7.9)	C: 2 Xyl1	H-2 Xyl1; H-5 Qui2
2	75.5 CH	3.91 t (7.9)	C: 1 Qui2	H-4 Qui2
3	75.1 CH	4.04 t (8.8)	C: 2, 4 Qui2	H-1 Qui2
4	**86.3** CH	3.56 t (8.8)	C: 1 Glc3; C: 5, 6 Qui2	H-1 Glc3; H-2 Qui2
5	71.3 CH	3.71 m		H-1, 3 Qui2
6	17.8 CH_3_	1.65 d (6.2)	C: 4, 5 Qui2	
Glc3 (1→4Qui2)				
1	104.6 CH	4.84 d (7.2)	C: 4 Qui2	H-4 Qui2; H-3, 5 Glc3
2	74.2 CH	3.91 t (7.9)	C: 1 Glc3	
3	77.3 CH	4.16 t (7.9)	C: 4 Glc3	H-1 Glc3
4	70.9 CH	3.98 t (7.9)		
5	77.7 CH	3.95 m		H-1, 3 Glc3
6	61.8 CH_2_	4.44 dd (2.0; 11.9)		
		4.11 dd (5.8; 11.2)		
Glc4 (1→4Xyl1)				
1	102.0 CH	4.87 d (7.9)	C: 4 Xyl1	H-4 Xyl1; H-5 Glc4
2	72.9 CH	3.82 t (8.4)	C: 1 Glc4	
3	**86.3** CH	4.10 t (8.4)	C: 1 MeGlc5; C: 4 Glc4	H-1 MeGlc5; H-1 Glc4
4	69.0 CH	3.86 m		
5	75.1 CH	4.05 m		H-1 Glc4
6	*67.1* CH_2_	4.98 m		
		4.69 m		
MeGlc5 (1→3Glc4)				
1	104.7 CH	5.12 d (7.8)	C: 3 Glc4	H-3 Glc4; H-3, 5 MeGlc5
2	74.3 CH	3.78 t (7.8)		H-4 MeGlc5
3	86.3 CH	3.62 t (7.8)	C: 2, 4 MeGlc5; OMe	H-1, 5 MeGlc5; OMe
4	69.7 CH	4.04 m	C: 3, 5 MeGlc5	
5	75.5 CH	3.99 m		H-1, 3 MeGlc5
6	*66.7* CH_2_	4.96 d (10.5)		
		4.79 dd (4.2; 10.5)		
OMe	60.2 CH_3_	3.75 s	C: 3 MeGlc5	

^a^ Recorded at 176.04 MHz in C_5_D_5_N/D_2_O (4/1). ^b^ Bold = interglycosidic positions. ^c^ Italic = sulfate position. ^d^ Recorded at 700.13 MHz in C_5_D_5_N/D_2_O (4/1). Multiplicity by 1D TOCSY.

**Table 7 marinedrugs-19-00187-t007:** ^13^C and ^1^H NMR chemical shifts, HMBC, and ROESY correlations of the aglycone moiety of kuriloside J (**7**) and K_1_ (**9**).

Position	δ_C_ mult. ^a^	δ_H_ mult. (*J* in Hz) ^b^	HMBC	ROESY
1	36.0 CH_2_	1.71 m		H-11, H-19
		1.33 m		H-3, H-5, H-11
2	26.6 CH_2_	2.10 m		
		1.86 m		H-30
3	88.5 CH	3.12 dd (4.2; 11.6)		H1-Xyl1
4	39.5 C			
5	52.6 CH	0.79 brd (11.8)		H-1, H-3, H-31
6	20.9 CH_2_	1.61 m		
		1.40 m		
7	27.8 CH_2_	1.51 m		
		1.18 m		H-5, H-32
8	41.1 CH	2.13 m		H-15, H-18, H-19
9	148.6 C			
10	39.0 C			
11	114.2 CH	5.19 m		H-1
12	35.8 CH_2_	2.01 m		H-17, H-32
		1.77 brdd (5.3; 16.2)		
13	45.2 C			
14	43.3 C			
15	43.9 CH_2_	2.15 m	C: 14	H-32
		1.36 m	C: 13	H-18
16	75.1 CH	5.76 m		H-32
17	56.2 CH	2.29 brt (7.9; 10.0)	C: 20	H-12, H-21, H-32
18	15.1 CH_3_	0.77 s	C: 12, 13, 14, 17	H-8, H-12, H-15, H-20
19	22.0 CH_3_	1.05 s	C: 1, 5, 9, 10	H-1, H-2, H-8, H-18
20	64.8 CH	4.28 dd (6.4; 10.0)		H-18, H-21
21	23.1 CH_3_	1.42 d (6.4)	C: 17, 20	H-12, H-17, H-18, H-20
30	16.4 CH_3_	1.01 s	C: 3, 4, 5, 31	H-2, H-6, H-31
31	27.8 CH_3_	1.16 s	C: 3, 4, 5, 30	H-3, H-5, H-6, H-30
32	18.8 CH_3_	0.67 s	C: 8, 13, 14, 15	H-7, H-17
OAc	21.1 CH_3_	2.17 s	OAc	
	171.2 C			

^a^ Recorded at 176.03 MHz in C_5_D_5_N/D_2_O (4/1). ^b^ Recorded at 700.00 MHz in C_5_D_5_N/D_2_O (4/1).

**Table 8 marinedrugs-19-00187-t008:** ^13^C and ^1^H NMR chemical shifts, HMBC, and ROESY correlations of the carbohydrate moiety of kurilosides K (**8**) and K_1_ (**9**).

Atom	δ_C_ mult. ^a,b,c^	δ_H_ mult. (*J* in Hz) ^d^	HMBC	ROESY
Xyl1 (1→C-3)				
1	104.7 CH	4.62 d (6.8)	C: 3	H-3; H-3, 5 Xyl1
2	**82.2** CH	3.88 t (7.6)	C: 1 Qui2; C: 1 Xyl1	H-1 Qui2
3	75.0 CH	4.07 t (7.6)	C: 2, 4 Xyl1	H-1 Xyl1
4	**78.9** CH	4.06 m		H-1 Glc4
5	63.4 CH_2_	4.32 dd (7.6; 11.4)		
		3.60 m		H-1 Xyl1
Qui2 (1→2Xyl1)				
1	104.5 CH	4.93 d (8.5)	C: 2 Xyl1	H-2 Xyl1; H-3, 5 Qui2
2	75.4 CH	3.84 t (8.5)	C: 1, 3 Qui2	H-4 Qui2
3	74.5 CH	3.93 t (8.5)	C: 2, 4 Qui2	H-5 Qui2
4	**86.9** CH	3.35 t (8.5)	C: 1 Glc3; C: 3, 5 Qui2	H-1 Glc3; H-2 Qui2
5	71.3 CH	3.65 m		H-1, 3 Qui2
6	17.7 CH_3_	1.60 d (5.3)	C: 4, 5 Qui2	
Glc3 (1→4Qui2)				
1	104.8 CH	4.69 d (6.5)	C: 4 Qui2	H-4 Qui2; H-3, 5 Glc3
2	74.1 CH	3.81 t (8.4)	C: 1, 3 Glc3	
3	76.9 CH	4.11 t (8.4)	C: 2, 4 Glc3	
4	70.8 CH	3.88 t (8.4)	C: 3, 5, 6 Glc3	
5	75.4 CH	4.10 m		H-1 Glc3
6	*67.6* CH_2_	5.08 brd (11.2)		
		4.64 dd (8.4; 11.2)	C: 5 Glc3	
Glc4 (1→4Xyl1)				
1	102.7 CH	4.84 d (7.5)	C: 4 Xyl1	H-4 Xyl1; H-3, 5 Glc4
2	73.2 CH	3.83 m	C: 1, 3 Glc4	
3	**86.0** CH	4.16 t (8.4)9	C: 1 MeGlc5; C: 2, 4 Glc4	H-1 MeGlc5
4	69.1 CH	3.83 m	C: 5, 6 Glc4	
5	74.8 CH	4.05 m		H-1 Glc4
6	*67.3* CH_2_	4.96 d (10.9)		
		4.61 m		
MeGlc5 (1→3Glc4)				
1	104.4 CH	5.19 d (7.2)	C: 3 Glc4	H-3 Glc4; H-3, 5 MeGlc5
2	75.1 CH	3.85 t (8.9)	C: 1, 3 MeGlc5	
3	86.9 CH	3.66 t (8.9)	C: 2, 4 MeGlc5; OMe	H-1 MeGlc5; OMe
4	70.2 CH	3.90 t (8.9)	C: 5 MeGlc5	
5	77.5 CH	3.89 m	C: 6 MeGlc5	H-1 MeGlc5
6	61.6 CH_2_	4.36 brd (12.5)		
		4.07 dd (5.4; 12.5)	C: 5 MeGlc5	
OMe	60.5 CH_3_	3.79 s	C: 3 MeGlc5	

^a^ Recorded at 176.04 MHz in C_5_D_5_N/D_2_O (4/1). ^b^ Bold = interglycosidic positions. ^c^ Italic = sulfate position. ^d^ Recorded at 700.13 MHz in C_5_D_5_N/D_2_O (4/1). Multiplicity by 1D TOCSY.

**Table 9 marinedrugs-19-00187-t009:** The cytotoxic activities of glycosides **1**–**9** and cladoloside C (positive control) against mouse erythrocytes, neuroblastoma Neuro 2a cells, and normal epithelial JB-6 cells.

Glycoside	ED_50_, µM	Cytotoxicity EC_50_, µM	
	Erythrocytes	JB-6	Neuro-2a
Kuriloside A_3_ (**1**)	>100.00	>100.00	>100.00
Kuriloside D_1_ (**2**)	>100.00	>100.00	>100.00
Kuriloside G (**3**)	76.26 ± 0.98	>100.00	>100.00
Kuriloside H (**4**)	6.85 ± 0.67	4.63 ± 0.08	38.28 ± 1.15
Kuriloside I (**5**)	>100.00	>100.00	>100.00
Kuriloside I_1_ (**6**)	10.34 ± 0.28	11.48 ± 1.02	56.63 ± 0.98
Kuriloside J (**7**)	47.61 ± 1.73	77.75 ± 0.27	>100.00
Kuriloside K (**8**)	>100.00	>100.00	>100.00
Kuriloside K_1_ (**9**)	20.97 ± 0.39	37.44 ± 0.13	>100.00
Cladoloside C (positive control)	0.54 ± 0.01	6.38 ± 0.08	9.54 ± 0.82

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
