# Peer review of "Triterpene Glycosides from the Far Eastern Sea Cucumber Thyonidium (=Duasmodactyla) kurilensis (Levin): The Structures, Cytotoxicities, and Biogenesis of Kurilosides A3, D1, G, H, I, I1, J, K, and K1"

_marinedrugs, 2021, doi:10.3390/md19040187_

Round 1
Reviewer 1 Report
See attached file

Author Response
The manuscript describes the isolation, structure elucidation, and examination of the cytotoxicity of nine new sulfated triterpene glycosides from the sea cucumber Thyonidium kurilensis. In addition, the structure of two more triterpene glycosides, which are obtained from a reaction of desulfation and therefore are not natural products, is described.
The structures of the new compounds are unexceptional, because the compounds differ only in some structural detail from known compounds form this and other species of sea cucumbers, and the same is true for their cytotoxicity; however, structure elucidation of these complex molecules is performed with competence, and all the structures are well supported by the spectroscopic data provided.
The structure of the compounds isolated so far from T. kurilensis are related into a reasonable, although speculative, biogenetic network. In this respect, Authors noted (line 460) that “there are some missing links (biosynthetic intermediates) in these biogenetic rows”. These missing intermediates could be looked for using feature-based molecular networking, as it has been done for the analysis of triterpene glycosides from the sea cucumber Holothuria atra (Grauso, L. et al. Mar. Drugs 2019, 17, 86, doi:10.3390/md17020086). This data analysis method is not difficult to implement, requires an LC-MS2 instrument that is available to Authors, and would allow a putative identification of minor glycosides in the extract without having to isolate them. Therefore, the molecular networking analysis could be added to this manuscript, or at least considered for the next papers in the series.
Unfortunately, the manuscript is often hard to read, because it contains many typos, grammar errors, and weird and involved sentences. An extensive revision of English language is absolutely necessary.
Apart from this, the paper could (and should) be improved in several ways, as detailed below:
- The abstract is too long (459 word, a maximum of 200 words is in indicated in the instructions for authors), so it should be dramatically
- Throughout the manuscript: Protons and carbons are identified as H-1, H-2... and C-1, C-2.... in Tables, but as H(1), H(2)... and C(1), C(2) in the text, the latter being a very unusual notation. Please use H-1, H-2.. and C-1, C-2..
- Abstract, line 11: The compounds reported may be “new”, but they are not “novel”, because they show little difference from known
- Figure 1: The current structural drawings do not allow a ready understanding of the structure of each molecule. At the very least, pentasaccharide and hexasaccharide chains should be drawn separately, and aglycone should be less close to each other. Further, Authors could denote each saccharide chain with a number (1, 2, ...) and each aglycone with a letter (a, b, ...), so that, for example, kuriloside A3 would be compound 1a. Alternatively, Authors could denote each saccharide chain with a capital letter (A, B, ...) and each aglycone with a small case letter (a, b, ...), and associate them to each name and structure number in a small
- Lines 85-93: this paragraph should be moves in the SI as a caption of Figure
- Line 95: Compound 1 is reported here as its sodium salt, but there is no demonstration that it was isolated as a sodium salt (ESI analysis cannot be a proof, elemental analysis would be necessary). On the other hand, this is not relevant, because the sodium counter-ion is not really part of the compound. The word sodium in the systematic names should be removed here and from all other systematic
- Lines 119-126: this paragraph should be moves in the SI as a caption of Figure S13, and so on for all the other
- Lines 142-149. This paragraph is so hard to read! Please use the abbreviations Xyl1, Qui2, Glc3, and so on that are used in Table 2. These abbreviations could be introduced in lines 140-141: “as one xylose (Xyl-1), one quinovose (Qui-2), two glucose (Glc3 and Glc5), and two 3-O-methylglucose (MeGlc4 and MeGlc6) residues”.
- Lines 154-155: “Thus, hexasaccharide disulfated chain of kuriloside G (3) is first found in the sea cucumber glycosides”. What about glycosides from other source?
- Lines 200-201: “This carbohydrate chain is first found in the glycosides of the sea cucumbers”. What about glycosides from other source?
- Line 249, line 323, and elsewhere: the terms “O-acetic group” or “O-acetate group” are incorrect. Use either “O-acetyl group” or “acetoxy group”.
- Lines 255-256: “has the aglycone with 16β,(20S)-dihydroxy-fragment that is unique in marine glycosides.” What about terrestrial glycosides?
- Line 333: the aglycone is new, but not
- Lines 436-447: this paragraph is formatted as a footnote of Table 9. Is this the case, or should it be normal text?
- Lines 713-810: the list of supplementary materials should be less detailed, and consequently much shorter.
Answers for Reviewer-1 comments:
- The abstract was shortened.
- Throughout the manuscript the designations for protons and carbons were changed to H-1, H-2... and C-1, C-2.....
- Abstract, line 11: The word “novel” replaced by the word “new” concerning compounds, aglycones or carbohydrate chains throughout the text.
- In accordance with the Reviewer comments, carbohydrate chains in the Figure 1 were denoted with the numbers I-VII and the aglycones with the letters A–E. They were associated with the numbers of compounds 1–9 in a table in Figure 1.
- Lines 85-93, Lines 119-126 and other paragraphs, discussing the fragmentation patterns of the studied compounds, corroborate their structures, established by NMR. So, we believe the moving of these paragraphs to the SI, as the captions of corresponding Figures, is inappropriate, due to the loss of valuable structural information for the readers.
- Line 95 and others about the sodium salts: At the beginning of studying of triterpene glycosides from the sea cucumbers the elemental analyses have been done with the atomic absorption spectroscopy demonstrating counter-ion of the sulfate groups is a sodium. Thus, the HR mass-spectra are appropriate method allowing to calculate brutto-formulae of the glycosides and all of them indicated the glycosides are sodium salts containing from one to three Na+ ions depending on the quantity of sulfate groups. Moreover, the loss of sulfate group in the MS/MS spectra is always observed as sodium salt. The protonated ions are rarely present in the mass-spectra of the glycosides that having no sulfate groups. So, the systematic names of the glycosides are right.
- Lines 142-149 and others: The abbreviations Xyl1, Qui2, Glc3, and so on, were used throughout the text for ease of reading.
- Lines 154-155, Lines 200-201: All carbohydrate chains found in the sea cucumber glycosides are compared with the corresponding moieties of other glycosides isolated earlier from different species of the sea cucumbers. Their comparison with the carbohydrate moieties of the glycosides found in plats or starfishes doesn’t look relevant due to the obvious significant differences of their structures, including the monosaccharide composition and the glycosidic bonds configuration and positions as well as the place of attachment to the aglycone moiety. So, the carbohydrate chains and aglycones composing the sea cucumber glycosides are “a priory” differ from these parts of the glycosides from other biological sources.
- Line 249, line 323, and throughout the text: the terms “O-acetic group” or “O-acetate group” were changed to “acetoxy group”.
- Lines 255-256: “has the aglycone with 16β,(20S)-dihydroxy-fragment that is unique in marine glycosides.” What about terrestrial glycosides? The senseless of comparison of the aglycone structures of the sea cucumber glycosides with those from plants is explained by the huge structural diversity of the triterpene skeletons derived from these sources. Structurally different triterpene nuclei are elaborated by diverse oxidosqualene cyclases (OSCs) using oxidosqualene as a substrate to produce carbocation that rearrangements lead to multicyclic triterpenes formation. There are no identical triterpene skeletons in the sea cucumbers and plants because the initial stages of their biosynthesis strongly differ. One of the main characteristic structural feature of the holothuroid’s aglycones is 9β-H configuration indicating 9β-lanosta-7,24-diene-3β-ol is a precursor of the aglycones having 7(8)-double bond. Similar plant triterpenoid metabolites are very rare and known and we are sure that all our compounds are unique both marine and terrestrial glycosides.
- Line 333: the aglycone is new, but not The inaccuracy is fixed.
- Lines 436-447: this paragraph is formatted as a footnote of Table 9. Is this the case, or should it be normal text? The text under the Table 9 was erroneously formatted as the footnote to the Table. Actually, it should be a normal text. The inaccuracy is fixed.
- Lines 713-810: The list of supplementary materials was shortened.
Reviewer 2 Report
Kalinin and coworkers described in this paper the final steps of extraction and purification (initial steps previously described) of sulfated saponins, kurilosides. Their exact structures were very well established thanks to 1D and 2D NMR analysis and MS spectra. Even if the description is somehow repetitive, this is clear and convincing on the both parts of the molecules, i.e. the aglycon and the glycosyl parts.
The authors also give some results related to biological activities. Unfortunately, the reason of the targets is absolutely not justified. All compounds can present cytotoxicity (or not!) according to the chosen model. Moreover, the ED50 and EC50 values are not compared to a reference product, so that the significance of the obtained values is not evident. Finally, the authors assert the membranolytic activity of the kurilosides. But what are the experimental proofs? So please improve significantly this chapter.
Concerning the supplementary data, the given spectra are of high quality.
In general, please also ensure homogeneity all around the text. This is indicated in the attached file, as well as other minor corrections. Provided that these remarks are positively considered, this study is available for publication in Marine Drugs.

Author Response
Comments and Suggestions for Authors
Kalinin and coworkers described in this paper the final steps of extraction and purification (initial steps previously described) of sulfated saponins, kurilosides. Their exact structures were very well established thanks to 1D and 2D NMR analysis and MS spectra. Even if the description is somehow repetitive, this is clear and convincing on the both parts of the molecules, i.e. the aglycon and the glycosyl parts.
The authors also give some results related to biological activities. Unfortunately, the reason of the targets is absolutely not justified. All compounds can present cytotoxicity (or not!) according to the chosen model. Moreover, the ED50 and EC50 values are not compared to a reference product, so that the significance of the obtained values is not evident. Finally, the authors assert the membranolytic activity of the kurilosides. But what are the experimental proofs? So please improve significantly this chapter.
Concerning the supplementary data, the given spectra are of high quality.
In general, please also ensure homogeneity all around the text. This is indicated in the attached file, as well as other minor corrections. Provided that these remarks are positively considered, this study is available for publication in Marine Drugs.
All other notes are in the marked text.
Answers for Reviewer-2 comments:
- In accordance with the Reviewer comments, formulae on the Figure1 were corrected. The carbohydrate chains were denoted with the numbers I-VII and the aglycones with the letters A–E. They were associated with the numbers of compounds 1–9 in a table on the Figure1.
- Tables 2 – 4, 6, 8: The numbers of sugar residues remain unchanged, but the corresponding numbers were changed in the formulae (Figure 1). So, they are brought into compliance to each other in the Figure 1, Tables and in the text.
- Line 313: “………C(6) of the glucose and 3-O-methylglucose residues…….” Replaced by “……. to C-6 Glc4 and C-6 MeGlc5……”
- The conditions of solvolytic desulfation are provided in Section 3.6 of “Materials and Methods”.
- The explanation justifying the choice of target cell lines: “Known earlier cladoloside C was used as positive control because it demonstrated strong hemolytic effect [23]. Erythrocytes are appropriate model for the studying of structure-activity relationships of the glycosides, since, despite many of them demonstrate a hemolytic activity, but the effect strongly depends on the structure of the compound. Normal epithelial JB-6 cells were used for searching the compounds, not cytotoxic against this cell line, but having selective activity against other cells. Triterpene glycosides of the sea cucumbers are known modulators of P2X receptors of immunocompetent cells when acting in nanomolar concentrations [24]. Neuroblastoma Neuro 2a cells are convenient model for the studying of agonists/antagonists of P2X receptors, the targets in the treatment of selected nervous system diseases. So, the activators, modulators and blockers of purinergic receptors are of great interest [4] and the compounds demonstrating high cytotoxicity against Neuro 2a cells could be more deeply studied with the models of neurodegenerative diseases.”
- It is known that the cytotoxicity of the glycosides is based on its selective bonding to the sterols of the cell membranes followed by the formation of complexes of the glycosides (mainly its aglycone part) with 5,6-unsaturated sterols of target cell membranes. These resulted in the disturbance of barrier properties of the membrane, pore formation and permeabilization of cells [Claereboudt, E.J.S.; Eeckhaut, I.; Lins, L.; Deleu, M. How different sterols contribute to saponin tolerant plasma membranes in sea cucumbers. Rep. 2018, 8, 10845, 1–11; Likhatskaya, G.N.; Yarovaya, T.P.; Rudnev, V.V.; Popov, A.M.; Anisimov, M.M.; Rovin, Yu.G. Formation of Complex of Triterpene Glycoside of Holothurin A with Cholesterol in Liposomal Membranes. Biofizika 1985, 30, 358-359.].
Reviewer 3 Report
This MS is too long and difficult to understand the presented data. The abstract is longer than the introduction. There is not enough background that can support the findings.
Overall, the quality and findings of this MS are not suitable for Marine Drugs.
Author Response
Comments and Suggestions for Authors
This MS is too long and difficult to understand the presented data. The abstract is longer than the introduction. There is not enough background that can support the findings.
Overall, the quality and findings of this MS are not suitable for Marine Drugs.
The abstract has been significantly shortened along with the referee note.
We strongly disagree concerning the article background supporting the findings. It seems to be enough and we are very glad that the other referees think so.
We also disagree with the general conclusion.
Reviewer 4 Report
This manuscript describes the extraction, purification and structural analysis of triterpenes glycosides from a Sea cucumber.
I recommend a major revision for these reasons:
- Line 52: The authors should explain why it’s a “reinvestigation”.
- Line 54: the isolated compounds are new ones ? (as explained in the abstract) If there are previously undescribed compounds, why the names like A3 or I1 or K1… It’s not clear for the reader.
- Line 68 and in the text: CHCl3 and H2O: 3 and 2 in subscript.
- Line 80: The structural elucidation of the aglycon of compound 1 is not detailed. Even if it’s a known aglycon, the authors have to explain how they have achieved the spectral analysis like the key signals or the absolute configuration of the hydroxyle in 16, the configuration of the lateral chain at the 17 position etc…In the formula, the configuration of the 17/20 linkage is not clear, and the H-5 is not correctly drawn. We can look at the table in the supplementary files but it’s not sufficient. The same thing with all the compounds even the knew ones. The authors explain the MS in details but the NMR analysis is too short.
- Line 94. The fragmentation isn’t sufficient for the identification of the sugars with the same mass (galactose/glucose, rhamnose/quinovose etc…) even the NMR data in the tables; how the authors justify the beta and D configuration? (same thing for the other molecules)
- Line 98: A comparison with a known compound is not sufficient
- Line 134: "In the HSQC spectrum" instead of "in the 1H and 13C NMR spectra"
- Line 137, please explain why beta configuration?
- Tables: I think there is a confusion between overlapped signals and multiplets. Multiplets are signals too complex to interpret easily. In the tables, “m” is used for a lot of signals. The multiplicities are really multiplets or the signals are overlapped without clear information? In this case, the authors should delete “m” where there are no multiplicity, and more multiplicities and coupling constants have to be added for aglycons and sugars signals.
- Line 252: There is another way to prove the 20(S) configuration, and not only based on biogenetic background?
- Why the authors used the erythrocytes test? All the saponins have a hemolytic activity…
- Why the structure/activity relationships are discussed in table 9 and not in the text?
- Figure 3: How the authors can be sure of this metabolic pathway?
- Line 674: What is the positive control?
- Line 689: same question
- Supporting information: the NMR spectra are not well resoluted, this is a scan version?
Author Response
This manuscript describes the extraction, purification and structural analysis of triterpenes glycosides from a Sea cucumber.
I recommend a major revision for these reasons:
- Line 52: The authors should explain why it’s a “reinvestigation”.
- Line 54: the isolated compounds are new ones ? (as explained in the abstract) If there are previously undescribed compounds, why the names like A3 or I1 or K1… It’s not clear for the reader.
- Line 68 and in the text: CHCl3 and H2O: 3 and 2 in subscript.
- Line 80: The structural elucidation of the aglycon of compound 1 is not detailed. Even if it’s a known aglycon, the authors have to explain how they have achieved the spectral analysis like the key signals or the absolute configuration of the hydroxyle in 16, the configuration of the lateral chain at the 17 position etc…In the formula, the configuration of the 17/20 linkage is not clear, and the H-5 is not correctly drawn. We can look at the table in the supplementary files but it’s not sufficient. The same thing with all the compounds even the knew ones. The authors explain the MS in details but the NMR analysis is too short.
- Line 94. The fragmentation isn’t sufficient for the identification of the sugars with the same mass (galactose/glucose, rhamnose/quinovose etc…) even the NMR data in the tables; how the authors justify the beta and D configuration? (same thing for the other molecules)
- Line 98: A comparison with a known compound is not sufficient
- Line 134: "In the HSQC spectrum" instead of "in the 1H and 13C NMR spectra"
- Line 137, please explain why beta configuration?
- Tables: I think there is a confusion between overlapped signals and multiplets. Multiplets are signals too complex to interpret easily. In the tables, “m” is used for a lot of signals. The multiplicities are really multiplets or the signals are overlapped without clear information? In this case, the authors should delete “m” where there are no multiplicity, and more multiplicities and coupling constants have to be added for aglycons and sugars signals.
- Line 252: There is another way to prove the 20(S) configuration, and not only based on biogenetic background?
- Why the authors used the erythrocytes test? All the saponins have a hemolytic activity…
- Why the structure/activity relationships are discussed in table 9 and not in the text?
- Figure 3: How the authors can be sure of this metabolic pathway?
- Line 674: What is the positive control?
- Line 689: same question
- Supporting information: the NMR spectra are not well resoluted, this is a scan version?
Comments for Reviewer 4.
- The word “reinvestigation” was chosen because the studying of kurilensis was begun in 80-s of XX century, when the complexity of glycoside composition of this species became obvious. Two glycosides were isolated that time due to the absence of the separation devices and approaches. Hence the attempt to find more new compounds was undertaken by us recently. The word “reinvestigation” was replaced by more appropriate “investigation” (Line 52).
- Historically the glycosides having the identical to each other carbohydrate moieties are considered as belonging to the same group and named by the same capital letter. So, kuriloside A3 has the oligosaccharide chain identical to earlier known kuriloside A and the aglycone identical to kuriloside F, but the combination of these parts in one molecule is found first making this compound new. Or the glycoside may have the new aglycone combined with the sugar chain that is the same for several compounds isolated from the glycosidic extract (this is the case of kuriloside D1).
- Line 68 and in the text: CHCl3 and H2O: 3 and 2 are in subscript.
- Line 80: The structural elucidation of 1 was added to the text: “The presence of five characteristic doublets at δH = 4.64–5.18 (J = 7.1–7.6 Hz), and corresponding to them signals of anomeric carbons at δC = 102.3–104.7 in the 1H and 13C NMR spectra of the carbohydrate part of 1 indicate the presence of a pentasaccharide chain and β-configurations of the glycosidic bonds. Monosaccharide composition of 1, established by the analysis of the 1H,1H-COSY, HSQC and 1D TOCSY spectra, includes one xylose, one quinovose, two glucoses and one 3-O-methylglucose residues. The signal of C(6) Glc4 was observed at δC = 67.1, due to α-shifting effect of a sulfate group at this position. The positions of interglycosidic linkages were established by the ROESY and HMBC spectra (Table S1). The analysis of NMR spectra of the aglycone part of 1 indicated the presence of 22,23,24,25,26,27-hexa-nor-lanostane aglycone with 16a-hydroxy,20-oxo-fragment and 9(11) double bond (Table S2) due to the characteristic signals: (δC0 (C-9) and 114.2 (C-11), δC = 71.1 (C-16) and δH = 5.40 (brt, J = 7.5 Hz, H-16), δC = 208.8 C-20)). The ROE correlations H(16)/H(15β) and H(16)/H(18) indicated a 16α-OH orientation in the aglycone of kuriloside A3 (1). 17αH-orientation, common for the sea cucumber glycosides was deduced from the ROE-correlation H-17/H-32”. The formulae have been improved.
- Line 94: D configurations of monosaccharide residues composing the glycosides of kurilensis were assigned earlier [Avilov, S.A.; Kalinovskii, A.I.; Stonik, V.A. Two new triterpene glycosides from the holothurian Duasmodactyla kurilensis. Chem. Nat. Compd. 1991. 27, 188–192.] as result of acid hydrolysis of the glycosidic sum followed by measurement of specific rotation of monosaccharides. Moreover, all known glycosides contain monosaccharides of D-series. Coupling constants JH-1/H-2 of each sugar unit are unambiguously indicate configuration of glycosidic bonds: β-glycosidic bonds characterized by J » 6.5 – 8.5 Hz, while α-glycosidic bonds have the constants J » 3 – 4 Hz [A.S. Shashkov, O.S. Chizhov, Bioorg Khim, 2 (1976) 437-497.]. All measured coupling constants in the studied compounds indicated β-configurations of all glycosidic linkages. Galactose and glucose are strongly differed by coupling patterns JH-4/H-5, and the first one has JH-4/H-5 » 8 – 9 Hz, typical for axial/axial protons. Rhamnose and quinovose are easily differentiated by the same manner and the presence of quinovose (not rhamnose) is confirmed by the ROE-correlations H-2Qui/H-4Qui observed in all the spectra of the glycosides of T. kurilensis. Moreover, since the glycosides form the sea cucumbers are for a long time studied compounds, their monosaccharide composition was investigated by chemical transformations, including the obtaining of peracetates of aldononitriles or the octyl-derivatives of monosaccharides followed by its GLC-analysis with the corresponding authentic samples. All these data allowed to conclude the monosaccharide composition of the glycosides is confined by Xyl, Qui, Glc and their methylated derivatives.
- Line 98: The discussion of elucidation of carbohydrate chain structure of kuriloside D1 was added: “Actually, six signals of anomeric doublets at dH = 4.70–5.28 (d, J = 7.5–8.2 Hz) and corresponding to them signals of anomeric carbons at dC = 103.7–105.7 indicated the presence of a hexasaccharide chain in kuriloside D1 (2). The presence of xylose, quinovose, three glucose and 3-O-methylglucose residues was deduced from the analysis of the 1H,1H-COSY, HSQC and 1D TOCSY spectra of 2. The positions of the interglycosidic linkages were elucidated based on the ROESY and HMBC correlations (Table S3). The presence in the 13C NMR spectrum of kuriloside D1 (2) of the only signal of O-methyl group at dC5 and the upfield shift of the signal of C-3Glc4 to dC 71.5 indicated the presence of non-methylated terminal glucose residue as the fourth unit in the chain.”
- Line 134: The sentence was clarified: “In the 1H and 13C NMR spectra of the carbohydrate part of kuriloside G (3), six characteristic doublets at δH65–5.19 (J = 7.0–8.1 Hz) and signals of anomeric carbons at δC 102.1–104.8, correlated with each anomeric proton by the HSQC spectrum, were indicative…..”
- Line 137, please explain why beta configuration? It was deduced from the coupling constants of anomeric protons J = 7.0–8.1 Hz, that were characteristic for β-configuration of the xylose, quinovose and glucose residues due to axial/axial coupling between H-1 and H-2 of each sugar residue.
- Comment concerning the multiplicity of the signals in NMR Tables: all the multiplicities and coupling constants that could be interpreted and calculated are provided in the Tables, the signals that marked as “m” are really multiplets. For example, in the Table 2 of carbohydrate chain of kuriloside G (3) the signal of H-4Xyl1 at δH 15 should be observed as “ddd”, due to the coupling with two protons of methylene group CH2-5Xyl1 as well as the proton of methyne group CH-3Xyl1, however due to their coupling constants are close, the signals are moved together and actually partly overlapped by the signal of H-3Xyl1. 1D TOCSY spectra allow analyzing each monosaccharide residue separately from the others as isolated spin system, so even the closely shifted signals can be differentiated and analyzed, except the cases when the signals indeed are multiplets. As for the signals of the aglycone protons, many of them have complicated coupling patterns due to the long-range coupling existing in polycyclic skeleton having double bonds. For example, the signal of H-8 should be “dd” due to the coupling with two protons at C-7, however H-8 have additional coupling constant with H-11 through π-electrons of 9(11)-double bond. So, the signal of H-8 is observed as “m”. The similar explanations are appropriate for other proton signals.
- Line 252: The assigning of C-20 configuration: (S)-configuration of C(20) stereo-center in nemogenin, the aglycone obtained earlier from the glycosidic sum of kurilensis, was established by the analysis of inter-atomic distances in the models of the (20R)- and (20S)-isomers and the NOE-experiments. The correlations H(17)/H(21), H(20)/H(18) observed in the ROESY spectrum of kuriloside I (5) as well as in the spectra of the other kurilosides and the closeness of the coupling constants J17/20 = 9.5 Hz (in the 1H NMR spectrum of 5), J17/20 = 10.0 Hz (in the 1H NMR spectrum of kuriloside J (7)), to that for nemogenin (J17/20 = 10.8 Hz) indicated the same (20S) configuration in the studied compounds. (20S)-configuration of holostane aglycones was established by X-ray analysis [Ilyin, S.G.; Reshetnyak, M.V.; Afiyatullov, S.S.; Stonik, V.A.; Elyakov, G.B. The crystal and molecular structure of diacetate of holost-8(9)-en-3α,16β-diol. Rep. USSR Acad. Sci. 1985, 284, 356–359., Ilyin, S.G.; Sharypov, V.F.; Stonik, V.A.; Antipin, M.Y.; Struchkov, Y.T.; Elyakov, G.B. The crystal and molecular structure of (23S)-acetoxy-9βH-holost-7-en-3β-ol and stereochemical peculiarities of the double bond migration from 7(8) to 8(9) and 9(11)-positions in holostane-type triterpenoids. Bioorg. Chem. 1991, 17, 1123–1128.].
The corresponding sentence in the manuscript was chaged: “(20S)-configuration in 5 was determined on the base of the closeness of the coupling constant J20/17 = 9.5 Hz to those in the spectra of kurilosides A1, C1 [19] and H (4) and corroborated by the observed ROE-correlations H(17)/H(21), H(20)/H(18) and biogenetic background.”
- Why the authors used the erythrocytes test? Actually, the majority of the glycosides have a hemolytic activity, but the effect strongly depends on the structure of the compound. So, structure-activity relationships are very well illustrated by the hemolytic action.
- Why the structure/activity relationships are discussed in table 9 and not in the text? The text under the Table 9 includes the discussion of structure-activity relationships. It was erroneously formatted as the footnote to the Table 9. The inaccuracy is fixed.
- How the authors can be sure of this metabolic pathway? The metabolic network is constructed on the base of real glycoside structures which were isolated from kurilensis, so, the majority of neighboring steps (precursor and subsequent compound) in the biosynthetic pathways (Figure 3) differ as results of one enzymatic reaction only (glycosylation, sulfation or methylation). Hence their comparative analysis clearly indicates the sequence of the biosynthesized molecules.
- What is the positive control? The phrase: “Known earlier cladoloside C was used as positive control because it demonstrated strong hemolytic effect [23].” and corresponding reference were added to the text. Line 674, Line 689: The phrase: “(including cladoloside C used as positive control)” was added.
- Supporting information: the NMR spectra are not well resoluted, this is a scan version? The NMR spectra are extracted from TopSpin software with using “Print with layout” function. The spectra are presented as 1-page for the COSY, HSQC, HMBC and ROESY in the “Supporting Materials” due to huge amount of data. In the process of the analysis of spectroscopic data the software TopSpin 3.0.b.8 allows scaling different regions of the spectra.
Round 2
Reviewer 2 Report
The authors propose a revised version of their manuscript. Improvments are significant and this version is suitable for publication.
Reviewer 4 Report
I think now this manuscript is in a suitable form to be published in Marine Drugs